# Quantifying the Uncertainty in the Eurasian Ice-Sheet Geometry at the Penultimate Glacial Maximum (Marine Isotope Stage 6)

Oliver G. Pollard[1], Natasha L.M. Barlow[1], Lauren Gregoire[1], Natalya Gomez[2], Víctor Cartelle[1,3], Jeremy C. Ely[4], and Lachlan C. Astfalck[5]

[1]School of Earth and Environment, University of Leeds, Leeds, UK
[2]McGill University, Montreal, QC, Canada
[3]Flanders Marine Institute (VLIZ), InnovOCean Site, Jacobstraat 1, Oostende, Belgium
[4]Department of Geography, University of Sheffield, Sheffield, UK
[5]Oceans Graduate School, The University of Western Australia, Perth, Australia

**Correspondence:** Oliver G. Pollard (o.g.pollard@leeds.ac.uk)

**Abstract.** The North Sea Last Interglacial sea level is sensitive to the fingerprint of mass loss from polar ice sheets. However, the signal is complicated by the influence of glacial isostatic adjustment driven by Penultimate Glacial Period ice-sheet changes, and yet these ice-sheet geometries remain significantly uncertain. Here, we produce new reconstructions of the Eurasian ice sheet during the Penultimate Glacial Maximum (PGM) by employing large ensemble experiments from a simple ice-sheet model that depends solely on basal shear stress, ice extent, and topography. To explore the range of uncertainty in possible ice geometries, we use a parameterised shear-stress map as input that has been developed to incorporate bedrock characteristics and the influence of ice-sheet basal processes. We perform Bayesian uncertainty quantification, utilising Gaussian Process emulation, to calibrate against global ice-sheet reconstructions of the last deglaciation and rule out combinations of input parameters that produce unrealistic ice sheets. The refined parameter space is then applied to the PGM to create an ensemble of constrained 3D Eurasian ice-sheet geometries. Our reconstructed PGM Eurasian ice-sheet volume is $48 \pm 8$ m sea-level equivalent (SLE). We find that the Barents-Kara Sea region displays both the largest mean volume and volume uncertainty of $24 \pm 8$ m SLE while the British-Irish sector's volume of $1.7 \pm 0.2$ m SLE is smallest. Our new workflow may be applied to other locations and periods where ice-sheet histories have limited empirical data.

## 1 Introduction

The Last Interglacial (LIG) (Marine Isotope Stage (MIS) 5e; 130-116 ka) was the last time in Earth's history that the Greenland and Antarctic ice sheets were smaller than today (Dutton et al., 2015), during a time when polar temperatures were 3-5 °C above pre-industrial values (Capron et al., 2014), raising global mean sea level by 5-10 m above present (IPCC, 2022). The timing, magnitude and spatial pattern of Last Interglacial sea-level changes are, in large part, caused by ice-mass changes during the interglacial as well as by those that occurred during the preceding glacial (MIS 6, 191-123 ka) cycle (Dendy et al., 2017; Rohling et al., 2008, 2019). The effect of ice-sheet melt on sea-level change is complex due to feedbacks between ocean water volume, perturbations of the Earth's rotational axis, Earth's gravitational field, and viscoelastic deformation of the solid

Earth due to changing ice and water loads (Milne and Mitrovica, 1998). Together, these processes are termed glacial isostatic adjustment (GIA) (Farrell and Clark, 1976; Mitrovica and Milne, 2003; Whitehouse, 2018), and form the primary drivers of spatially variable relative sea-level (RSL) change on glacial-interglacial timescales.

Regional LIG RSL changes are a consequence of the distribution and timing of terrestrial ice-mass deglaciation during the preceding glacial. Last Deglaciation ice-sheet histories included in GIA reconstructions are well constrained by a wealth of geological data (Clark and Mix, 2002; Dalton et al., 2020; Hughes et al., 2016) and tested against comprehensive RSL databases (e.g., Peltier, 2004; Shennan et al., 2006; Stuhne and Peltier, 2017; Tarasov et al., 2012). In contrast, for glacial periods prior to the Last Glacial Maximum (LGM), including the Penultimate Deglaciation (typically correlated to the end of

MIS 6, and regionally in Europe, the late Saalian glacial phase) that preceded the LIG, a paucity of geomorphological and chronological constraints for ice extent, thickness, and volume means that older ice-sheet reconstructions are much harder to constrain. This presents a significant source of uncertainty for studies that focus on ice and water loading changes during the LIG (Barlow et al., 2018; Düsterhus et al., 2016). One notable uncertainty in the Penultimate Glacial Period ice history is the Eurasian ice sheet, as its extent was thought to have been significantly larger during the PGM than the LGM (Batchelor et al.,

2019; Svendsen et al., 2004) (Figure 1). Geological data suggests that the preceding glacial Eurasian ice sheet was typified by more than one period of ice advance during late Saalian. In western Europe two significant phases of ice advance occurred; the Drenthe (ca. 175-160 ka) which extended south of the LGM ice extent in the Netherlands, and the latter Warthe readvance (ca. 150-140 ka) which terminated within the limits of the Drenthe glacial maximum (Toucanne et al., 2009; Ehlers et al., 2011; Ehlers and Gibbard, 2004). To the east, a period of Saalian ice advance in central Russia deposited the extensive Moscow

till, which is now commonly ascribed to MIS 6 (Shik, 2014), though chronological uncertainties means it remains unresolved how this glacial deposition correlates to the advance/retreat phases in the west. It is reasonable to assume that the Penultimate Deglaciation of the Eurasian ice sheet may have been asynchronous, as it was during the Last Deglaciation (Patton et al., 2017), with parts of the ice sheet reaching its maximum position at the same time as other areas retreated. This difference in timing and extent would result in a differing pattern of solid Earth displacement and RSL change during the LIG, in both the near

and far-field, compared to the Holocene (Cohen et al., 2022; Dendy et al., 2017; Lambeck et al., 2006; Rohling et al., 2008). However, to better constrain this, more chronological data is needed to reconstruct the spatially variable timing and extent of the ice load during the Penultimate Glacial Period across Europe (Lauer and Weiss, 2018).

    Previous work reconstructing the configuration of the Eurasian ice sheet has primarily focused on the Last Deglaciation (Clark et al., 2022; Gowan et al., 2021; Patton et al., 2016; Peltier et al., 2015; Tarasov et al., 2012) with some notable excep-

tions extending to the Penultimate Deglaciation (Colleoni et al., 2016; Lambeck et al., 2006). Ice-sheet reconstructions can be categorised as either 2D, which aim to outline the ice sheet extent, or 3D, where the geometry (thickness and extent) of the ice sheet is estimated. Detailed 2D reconstructions of the Last Deglaciation have been compiled from available geomorphological constraints describing the full chronological evolution of the ice sheet at high temporal resolutions of up to 0.5 ka (Batchelor et al., 2019; Hughes et al., 2016). In contrast, 2D reconstruction efforts for the Penultimate Deglaciation Eurasian ice sheet are

more limited and have focused on the maximum asynchronous ice limit during the Penultimate Glacial Cycle, since intermediary deglaciation margins are difficult to constrain and date with the available geomorphological evidence (Batchelor et al.,

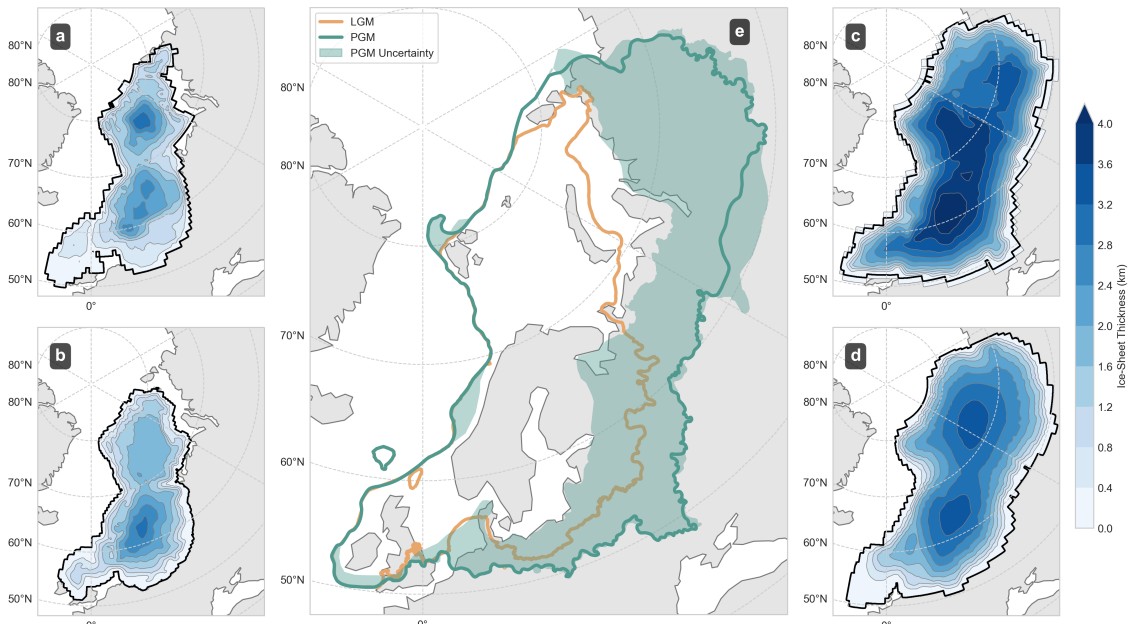

**Figure 1.** LGM and PGM Eurasian Ice Sheet reconstructions: (a) 26 ka ICE-6G (Peltier et al., 2015) and (b) 21 ka GLAC-1D (Tarasov et al., 2012) reconstructed Eurasian ice-sheet thickness at their respective maximum Eurasian ice-volume configurations during the Last Glacial Maximum. PGM maximum ( 140 ka) Eurasian ice-sheet thickness from (c) Colleoni (2009) and (d) Lambeck et al. (2006). (e) Comparison of ice margins from Batchelor et al. (2019), with the green band showing the area between the MIS 6 maximum and MIS 8 best-estimate margins.

2019; Svendsen et al., 2004). 2D reconstructions are limited in their application to GIA modelling since they do not provide ice thickness information.

Three main approaches have been employed to estimate 3D Eurasian ice-sheet geometry, and therefore ice thickness and
volume: GIA inversion, dynamic ice-sheet modelling, and simple ice-sheet modelling. In the first, solutions to the inverse GIA problem are calculated by tuning a combination of global ice reconstruction, radially varying Earth viscosity, and lithospheric thickness to fit a global set of RSL records and modern Global Navigation Satellite System (GNSS) data. This method has been applied in the generation of numerous Eurasian ice-sheet reconstructions during the Last Deglaciation (Lambeck et al., 2006; Peltier, 2004; Peltier et al., 2015) while also having been applied to the Penultimate Deglaciation (Lambeck et al., 2006). By
design, GIA inversion ice-sheet load solutions are consistent with empirical constraints on rebound and sea-level data, when combined with the corresponding adopted viscoelastic Earth structure, but do not ensure physically consistency with known ice-sheet physics, often leading to physically implausible reconstructions.

In the second approach, 3D thermodynamic ice-sheet models, driven by climate forcing, are used to model 3D time-evolving ice-sheet geometry. This approach has been applied to the PGM in combination with a prescribed climate forcing to produce a
3D Eurasian ice-sheet reconstruction that, at equilibrium, matches the Svendsen et al. (2004) Eurasian ice margins (Colleoni,

2009; Colleoni et al., 2016; Peyaud, 2006). In turn, this reconstruction has been used to drive ice-sheet sensitivity experiments (Wekerle et al., 2016). Similarly, Abe-Ouchi et al. (2007) used a dynamic ice-sheet model, driven by a general circulation model, to model Northern Hemisphere ice sheets over late Quaternary glacial cycles, which are used as boundary conditions for transient climate simulations of PMIP4 (Menviel et al., 2019). In other work, thermodynamical climate-driven ice sheet simulations have been performed by Tarasov et al. (2012) and Patton et al. (2017), nudged to fit constraints from 2D reconstructions, near-field sea level data, meltwater history, and climate evolution. While dynamic models ensure more physically plausible ice-sheet geometries they are also dependent on the reliability of the climate-forcing used.

Finally, the simple ice-sheet model approach is designed to generate ice geometries that approximate the profile of a steady-state ice-sheet for a given margin. This technique has been used in both regional reconstructions, such as the Last Deglaciation of the western Laurentide (Gowan et al., 2016b), as well as global ice-sheet margins (Gowan et al., 2021) during the Last Deglaciation.

The large uncertainties and limited data available from which to constrain the pattern and timing of the Penultimate Deglaciation of the Eurasian ice sheet (Rohling et al., 2017; Ehlers et al., 2011; Hughes et al., 2011) means it must be tackled with robust and efficient methods of uncertainty quantification and parameter sampling for the problem to be tractable (Andrianakis et al., 2015; Astfalck et al., 2021; Williamson et al., 2013). LGM studies show it is possible to use uncertainty quantification techniques, combined with 3D dynamic ice-sheet modelling, to estimate a range of plausible ice-sheet histories (Gregoire et al., 2016; Tarasov et al., 2012; Gandy et al., 2021). However, reliance on poorly constrained rebound data required for GIA inversion modelling (Lambeck et al., 2006) or assumptions of highly uncertain climate data used in dynamic ice-sheet simulations (Abe-Ouchi et al., 2007; Peyaud, 2006) make these approaches challenging to constrain for the Penultimate Deglaciation and give only a very limited view of possible pasts with no grasp on the vast range of plausibility. In addition, computational requirements make quantification of uncertainties intractable if the models used are too complex. Therefore, the fast execution speeds and small number of input parameters make simple ice-sheet modelling a well suited approach for tackling the challenges of the PGM within a Bayesian uncertainty quantification framework.

In this paper, we develop a new technique to generate plausible Eurasian ice-sheet geometries for the PGM where we have little information on ice thickness and dynamics, accounting for uncertainty, and provide an ensemble of ice sheets that have been systematically tested. We utilise ICESHEET, a simple ice-sheet model whose minimal input requirements enables the production of large ensemble simulations with controlled sources of uncertainty (Gowan et al., 2016a). We demonstrate how the two-dimensional, uncertain shear stress input to the model can be parameterised and systematically varied to produce an ensemble of physically consistent ice-sheet geometries. We then test and calibrate the model and input shear stress map on the Last Deglaciation to rule out implausible input parameters and produce a new simulation of the Eurasian Last Deglaciation in the process. Finally, we apply the information gained from this process to produce ensembles of ice-sheet geometries for the PGM that can serve as input to subsequent GIA modelling to robustly quantify uncertainties.

## 2 Models and Methods

### 2.1 ICESHEET Simulator

ICESHEET is an ice-sheet model (Gowan et al., 2016a) that assumes steady-state conditions and a simple, perfectly plastic ice-sheet rheology to rapidly generate physically plausible ice-sheet reconstructions from only three 2D model inputs: ice-sheet margins, regional topography, and basal shear stress (based upon the physics first developed by Nye (1952), Reeh (1982), and Fisher et al. (1985)). Using an iterative process, ICESHEET calculates thickness profiles along flowlines that are generated at regular intervals within the prescribed ice margin. Flowline positions, and thus the ice-sheet thickness profile, are dependent

on the 2D input topography and shear stress maps (Gowan et al., 2016b). The shear stress map serves as a tuning input that can be calibrated or inverted to produce a target ice-sheet geometry, though significant uncertainties exist in determining basal shear stress (Sect. 2.2).

The model has been applied where large uncertainty in inputs required for dynamic ice-sheet models, such as climate, have reduced the confidence in using the outputs of such models as inputs to sea-level models due to misfits against ice extent

and volume distributions that impact GIA, and where large numbers of runs are required making computational efficiency paramount, such as in the exploration of variable global ice-sheet configurations (Gowan et al., 2021). Limited constraints on climatic conditions, the requirement for large ensemble simulations to explore the range of plausible scenarios, and a need for well-defined sources of uncertainty make ICESHEET an ideal choice for exploring uncertainty in ice sheet configurations during the PGM.

Two model parameters determine the resolution of a reconstruction with ICESHEET: contour elevation interval and flowline spacing. For our reconstructions, we use values of 20 m and 5000 m respectively in order to balance compute time with resolution. The 2D model inputs are defined on a Lambert Azimuthal Equal Area (LAEA) projection centred on longitude 0, latitude 90, using the WGS84 ellipsoid, and with boundaries defined at -1265453 m to 4159547 m in the x direction and -4722734.8 m to 1352265.2 m in the y direction with no x or y offsets, covering the Eurasian region at a resolution of 5 km.

In the following subsections, we describe the setup and inputs to simulations of the Last Deglaciation and PGM.

### 2.2 Uncertainty Quantification

ICESHEET, owing to the large uncertainties in the shear stress input, is capable of producing a wide range of ice-sheet geometries for both the Last Deglaciation and the PGM. While it is useful to retain some of this possible set of geometries for the purpose of uncertainty quantification, not all simulations will fall within our expectations of plausible Eurasian configura-

tions. Existing GIA reconstructions provide constraints on ice-sheet thickness during the Last Deglaciation and it is desirable to transpose this information to the PGM through model calibration. Bayesian uncertainty quantification techniques exist to explore uncertainty and calibrate physical models (Astfalck et al., 2021). However, because ICESHEET's primary input is the two-dimensional (2D), extremely heterogeneous, and poorly constrained basal shear stress matrix, "out-of-the-box" methods for sampling uncertain model inputs are unsuitable. Moreover, due to the major simplifications applied within ICESHEET, this

2D input should not only represent ice basal shear stress linked with bedrock geology, but should also encompass the effect

of missing ice surface mass balance and the influence of basal processes. Thus, a bespoke framework for quantifying past ice-sheet uncertainty with simple ice-sheet models such as ICESHEET is needed.

We first employ ICESHEET to produce a new simulated history of the Last Deglaciation that we then calibrate against independently derived, regionally aggregated volume metrics for the Last Deglaciation by employing a Bayesian uncertainty
quantification method called history matching. History matching allows us to identify regions of the ICESHEET input parameter space for which ICESHEET simulations are able to match the regional volume estimates that are expressed in published reconstructions (used here as an "observation") given the uncertainty in the model and target data (Williamson et al., 2015). This space is referred to as the Not Ruled Out Yet (NROY) space and, once identified using the Last Deglaciation constraints, can then also be applied to refine our set of reconstructions for the PGM where empirical constraints on published models are far
more limited. This procedure also allows us to identify systematic difference between the geometry simulated by ICESHEET and those reconstructed through GIA modelling, thus testing the capability of our modelling approach in providing meaningful ice geometries for use in sea-level and climate simulators.

## 2.3 Model Setup for the Last Deglaciation

We consider two spatiotemporal reconstructions of Eurasian ice-sheet thickness and regional topography during the last glacial
period: GLAC-1D (Tarasov et al., 2012) and ICE-6G (Peltier et al., 2015). These reconstructions have been selected as they are widely used, well regarded, and more accessible than others (Ivanovic et al., 2016; Lambeck et al., 2006; Menviel et al., 2019) while also representing two contrasting modelling methodologies that are both independent to the ICESHEET methodology (Gowan et al., 2021). GLAC-1D is the result of a large ensemble of thermodynamic ice-sheet simulations driven by climate reconstructions that have been nudged and selectively refined to fit relative sea-level records. It is provided every 0.1 ka at a
spatial resolution of 0.25° latitude and 0.5° longitude (Tarasov et al., 2012). ICE-6G is a solution to the inverse GIA problem and is provided at 0.5 ka after 21 ka, and 1.0 ka before, with spatial resolution of 1° latitude and longitude (Peltier et al., 2015). ICE-6G provides a better fit to sea-level records than GLAC-1D, but the ice geometry is not compatible with ice-sheet physics (Stuhne and Peltier, 2017), while GLAC-1D provides glaciological consistent ice-sheet geometries that account for ice-flow physics and climate forcing (Tarasov and Peltier, 2002). Both reconstructions account for GIA effects, provide accompanying
topography inputs, match against RSL data, and include a range of time slices that span the full deglaciation.

We extract the ice margin from each Last Deglaciation reconstruction, for use as input to ICESHEET, to ensure that we are able to accurately compare difference between thickness slices generated by ICESHEET and those of the reconstruction considered. To do this, we reproject and interpolate each reconstruction onto the same model grid as ICESHEET before applying an algorithm that produces ice-margin geometries from the gridded thickness data (Appendix A).
When using ICESHEET to simulate past ice sheets, the input topography needs to be adjusted for GIA. Since our aim is to reproduce ICE-6G and GLAC-1D volumes, we simply use the topography deformation fields provided by each model, reprojected onto our model grid. We run the ICESHEET model with topography and margins from GLAC-1D and ICE-6G, at 22, 20, 18 and 16 ka. These times are chosen since they capture a range of ice sheet deglaciation thickness and extent configurations while excluding the very thick slices >22 ka, which are poorly constrained by sea-level data, and those of small

extent after 16 ka which are less relevant for producing the extensive PGM. We label these simulations ICESHEET$_{1D}$ and
ICESHEET$_{6G}$.

## 2.4 Model Setup for the Penultimate Glacial Maximum

### 2.4.1 Ice Sheet Margin

We generate a range of possible ice-sheet margins based on the late Quaternary ice extent maps produced by Batchelor et al.
(2019) derived from a compilation of empirical and modelling evidence, which for MIS 6 includes 25 empirical extent outlines,
40 empirical point-source datapoints, and 5 modelled ice extents. Batchelor et al. (2019) produce minimum, best-estimate, and
maximum extent margins for MIS 6 which primarily differ in extent in Siberia (Figure 1). In this work we select three margins
in order to explore the uncertainty in the PGM configuration of the Eurasian ice sheet (Figure 1). We use the MIS 6 best-
estimate margin noting that this represents the maximum extent the ice sheet would have reached at any one time between ca.
190-132 ka; though in the west most likely corresponds with the Drenthe stage (>150 ka) given the extensive southern ice sheet
position in western Europe and the North Sea. We also utilise the MIS 6 maximum margin to explore the uncertainties in the
maximum Siberian extent. Given the potential for a smaller ice sheet during the latter part of the Saalian complex (which is
not captured in Batchelor et al.'s minimum MIS 6 margin) we use their MIS 8 best estimate map as a proxy for a late Saalian
ice extent where the maximum ice position in western Europe was further to the north during the Warthe substage (<150 ka).
This provides a starting point by which to explore the uncertainty in the PGM configuration, that can only be furthered with
improved temporal and spatial constraints.

Margin extent is included as a continuous parameter in our experimental design that varies between 0 and 1, where a values
of 0, 0.5, and 1 correspond to the minimum (MIS 8 best-estimate), most likely (MIS 6 best-estimate), and maximum (MIS
6 maximum) extents respectively. Values that fall between these points represent intermediary margins between the three
configurations which we generate by employing a novel shape-interpolation algorithm we have developed for this purpose.
Since the Batchelor et al. (2019) MIS 6 best-estimate reconstruction is restricted to the subset of their data that they judge to
have the highest reliability, we apply a normal probability distribution to our margin extent parameter, centred around 0.5, to
ensure that margins closest to this best-estimate are most common in our ensemble.

### 2.4.2 Topography

For simulations of the Last Deglaciation, we employ pre-existing models of topography changes due to GIA as provided
with the adopted GLAC-1D and ICE-6G ice histories for use as input to ICESHEET. For the PGM no such pre-existing GIA
deformation model exists for our ice load and yet GIA driven changes in topography beneath the ice sheet play an important
role in determining ice-sheet geometry, contributing up to a 20% increase in total ice volume over the Penultimate Glacial
cycle relative to a simulation where topography remains fixed (Gowan, 2014). In order to account for GIA, we first estimate
the topographic deformation field that would result from the solid Earth underneath the Eurasian ice sheet being at (or close to)
isostatic equilibrium with the ice load. To estimate this fully compensated topography associated with a given load, we adopt

the fully relaxed form of the simple Elastic Lithosphere Relaxing Asthenosphere (ELRA) model (Huybrechts and Wolde, 1999):

$$w_q(r) = \frac{qAL^2}{2\pi D}\text{kei}\left(\frac{r}{L}\right)$$

$\quad q = \rho_i g h$

$$L = \left(\frac{D}{\rho_b g}\right)^{\frac{1}{4}}$$

where $h$ is the thickness of the ice, $g$ is the acceleration due to gravity ($9.81 \text{ ms}^{-2}$), $\rho_i$ is the density of ice ($916 \text{ kgm}^{-3}$), $q$ is the applied ice load, $w_q$ is the solid earth response to loading at a radial distance $r$ from the load, $A$ is the area of an applied load cell, $L$ is the flexural rigidity length scale, $\rho_b$ is the bedrock density ($3300 \text{ kgm}^{-3}$), and $D$ is the flexural rigidity of the 210 lithosphere ($1025 \text{ Nm}$).

The assumption of full compensation could be considered reasonable, given the lack of constraints during this time, if the ice-sheet maximum configuration endured for a sufficiently long duration. However, in order to account for the possibility of partial deformation, we include a continuous scaling parameter in our ensemble that scales the fully relaxed deformation field, ranging between 0.475 and 1, for a given ice-sheet load such that lower values result in a smaller magnitude of deformation. 215 The lower bound of this parameter is constrained by comparing the (partially relaxed) topography at 20ka predicted in the GLAC-1D model to a calculation of the fully compensated topography that would result from inputting the GLAC-1D ice cover at 20ka and modern topography into the equations above.

In order to approximate topography deformation at the PGM, we begin by reprojecting the RTopo-2 modern day global topography (Schaffer et al., 2016), originally provided at a 0.5 degree resolution in latitude-longitude coordinates, onto the 220 LAEA model domain, interpolating onto our chosen Eurasian grid at a 5 km spacing, and applying a $1\sigma$ Gaussian Blur in order to smooth any sudden changes in elevation. This smoothing is required because ICESHEET can fail to run if large topography gradients are present when calculating flowline shapes. Following the approach of Gowan et al. (2021), we run ICESHEET with this modern-day topography to calculate an initial ice-sheet thickness which is then used as the load input to the ELRA model in order to calculate the resulting deformed topography. This new deformed topography is then scaled by 225 the topography parameter before being used as input in a second iteration run of ICESHEET in order to calculate the resulting ice-sheet thickness.

## 3 Parameterising the Shear Stress Input Map

The primary control and biggest source of uncertainty in ICESHEET is the 2D input shear stress map. The presence, composition, and thickness of deformable sediments underneath an ice sheet impacts the friction at the ice-bed interface, which, in turn, 230 affects the flow of ice and thus the local ice-sheet thickness and geometry. Nye (1952) originally related these quantities by balancing the shear stress at the base of the ice sheet with the driving stress which, after expansion by Reeh (1982) and Fisher

et al. (1985), was modified to include the impact of topography. Studies employing this theory have used surface geology data to develop maps of shear stress (Fisher et al., 1985; Gowan et al., 2021, 2016b).

The shear stress values can be calibrated or inverted to match a target ice geometry or varied to predict a range of plausible geometries. However, random sampling of such 2D inputs within the context a Bayesian uncertainty quantification framework presents a significant challenge since the number of independent parameters likely make experiments computationally unfeasible. To simplify this problem, studies typically employ one of two approaches to deal with 2D inputs: random error field generation, or parameter decomposition. In the first approach, each value within the 2D input is modelled as having an error described by a probability density function and spatial autocorrelation which, together, allows for the random generation of 2D error fields. When summed with the original values, error fields represent possible realisations of the 2D input (Zhao and Kowalski, 2020). Alternatively, the approach of parameter decomposition aims to reduce the number of parameters by collecting groups values with similar properties that together could be assumed represent spatial collections of homogeneous behaviour, and that can therefore be varied as a single parameter.

In a similar manner to parameter decomposition, previous studies have divided their study area into a set of geographic regions that are each assumed to have the same internal average shear stress value (Gowan et al., 2021, 2016b). The shear stress values are chosen to reflect a combination of known accumulation rates, with lower values used for areas that have higher moisture scarcity; evidence of ice thickness including GPS uplift rates; knowledge of underlying sediments which inform the deformability of the bed; topographic elevation (Fisher et al., 1985; Gowan et al., 2016b; Reeh, 1982); and, in some cases, modified in order to fit a database of known RSL data (Gowan et al., 2021, 2016b). However, this approach still produces a complex mosaic of independent regions that are too numerous to incorporate into a Bayesian uncertainty framework. To overcome this, we also decompose our study area into geographic regions of similar shear stress, derived from geological maps and satellite data but we choose not to follow the approach of previous work in converging on a single tuned shear stress input. This is because, firstly, such an approach results in a single 'best-fit' ice-sheet simulation output and, secondly, lacks the possibility of rigorous uncertainty quantification since such analysis with many independently varying shear stress regions becomes intractable. Therefore, we instead opt to incorporate the uncertainty inherent in the shear stress values of similar regions, enabling the production of a range of ice-sheet simulations by propagating uncertainty through our ensemble.

## 3.1 Sediment Distribution

In this paper, we adapt a basal shear stress map, developed for Eurasia during the Last Deglaciation, utilised in Gandy et al. (2018) and Clark et al. (2022). This map was constructed by dividing the bed of the Eurasian ice sheet into distinct surface geological and geomorphological units, in consultation with geological mapping, sediment thickness maps, and the distribution of glacial landforms observed by satellite imagery and digital elevation models. In the original map, five landscape categories were distinguished: i) palaeo-ice streams; ii) marine sediments; iii) thick and iv) thin terrestrial Quaternary sediments, as indicated by subglacial bedforms and on sediment maps; and v) exposed bedrock surface (Gandy et al., 2018). Due to uncertainties in the identification of such sediment categories during the PGM, we modify the original map in four ways to make it applicable to modelling the Penultimate Deglacial history and to keep our quantification of uncertainties tractable: (i) merging the original

continuous sediment and discontinuous sediment categories into a single 'onshore' category to reflect the lack of evidence to constrain the location of regions of discontinuous sediment during the Penultimate Glacial Period; (ii) defining the underlying sediment type for each ice streaming region so that their length may be altered and the underlying sediment revealed (Sect. 3.2); (iii) adding additional ice streaming regions in the Eastern sector and creating a separate ice streaming layer for the PGM (Sect. 3.3); (iv) expanding the Southerly and Easterly extent of the map to encompass the greater Eurasian PGM ice-sheet extent. Regions within our adapted map are therefore categorised by their underlying sediment in the absence of ice streaming (Figure 2a) as well as their potential to ice stream during the PGM and Last Deglaciation (Figure 2b).

Each category has an associated shear stress value uncertainty range, derived from our expert judgement, and described in Table 1. The larger extent of the Eurasian ice sheet during the Penultimate Glacial Period means that we require a shear stress map that extends further south, into Continental Europe, and further east towards Siberia. We designed these additional regions based on a digitally compiled maps of geology (Persits et al., 1997) alongside modern satellite imagery. Our final shear stress map for Eurasia for the Penultimate Glacial Period consists of 740 categorised regions (Figure 2).

## 3.2 Ice Streaming

Ice streams are corridors of fast-flowing ice that occur towards the exteriors of ice sheets and significantly reduce local ice thickness (Stokes and Clark, 2001). It is important that these regions are represented explicitly in ICESHEET, as the model lacks the dynamic mechanisms needed to generate ice streams on its own (Hindmarsh, 2009; Gandy et al., 2019), and so they are instead included as areas of very low shear stress. This enables ICESHEET to capture their main effect for GIA models, to reduce overall ice thickness. Evidence for the configuration of historic ice streaming relies on the identification of flow patterns, shapes, and deformed bed conditions within the geomorphological record (Stokes and Clark, 2001, 1999). Ice stream margin features can be dated (e.g. radiocarbon, cosmogenic nuclide, or optically stimulated luminescence) in order to infer the time the associated ice stream was active (Bentley et al., 2010; Stokes et al., 2015). Identifying and dating ice-streaming regions during the Penultimate Glacial Period poses a greater challenge compared to the Last Glacial Period as the period pre-dates the application of 14C methods and much of the sediment left behind by streaming has been removed by subsequent glacial activity. This is especially true for the southern margin of the Eurasian ice sheet (Joon et al., 1990; Laban, 1995; Sokołowski et al., 2021). By comparison, the extent-limiting influence of the continental shelf break and topography of troughs on the shelf increases confidence that ice streaming in the northern Penultimate Glacial Period Eurasian ice sheet was similar to the Last Glacial Period.

We represented ice streams in our shear stress map with a low shear stress value. To reflect the differences in streaming configurations as well as the disparity in geospatial constraints between the two glacial periods, we produce two separate maps of ice streaming during the Last Glacial Period and the Penultimate Glacial Period. The Penultimate Glacial Period layer is identical to the Last Glacial Period layer in the north, except for the addition of two streaming regions in the northeast, since the Eurasian ice sheet reached a similar extent during both glaciations (Figure 1). However, we completely remove ice streaming along the southern region in the PGM sheet stress map since evidence constraining streaming positions during the Penultimate

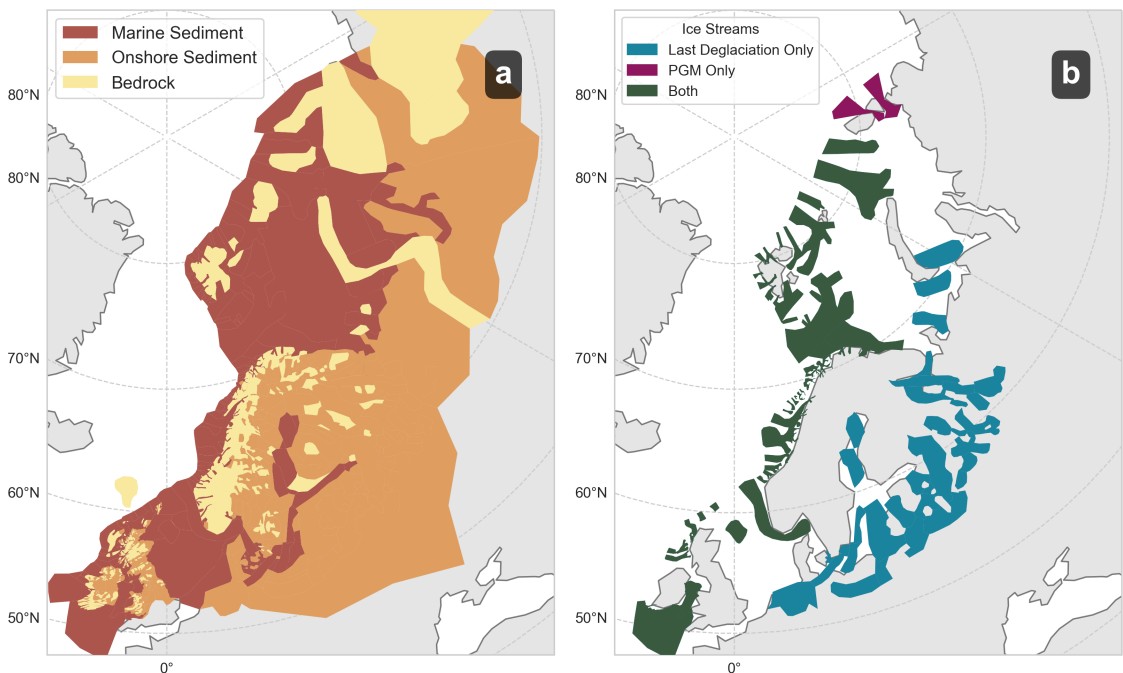

**Figure 2.** Components of the Shear Stress Map: (a) Underlying sediment category, which is used in the absence of overlying basal modification, shown for each region. (b) Regions capable of ice streaming, which is represented in our model by a low shear stress, for the Last Deglaciation (blue), PGM (purple), and both (green).

Glacial Period are sparse, while the larger extent of the PGM ice sheet (Figure 1) means the mapped Last Glacial Period ice streams do not apply as they would terminate within the interior of the ice sheet.

### 3.3 Ice Sheet Influence on Basal Conditions

Prescribing shear stress values based on geological surface type ignores the influence of basal conditions on sliding (Tsai et al., 2015; Weertman, 1957). However, the basal conditions can influence the effective shear stress and, in turn, affect the geometry of an ice sheet. In order to better capture ice-sheet basal interactions we account for the influence of three such effects: cold based ice, active ice streaming, and hybrid ice streaming. The first approximates the effects on basal conditions when ice becomes frozen to the surface in the central interiors of large ice sheets (Bierman et al., 2015). Cold-based ice has a high effective shear stress whether the bed is made of hard bedrock or soft sediment. The cold-based ice modification introduces this idea through two parameters. The first controls the size of the cold-based region (modelled as distance of unfrozen region from the margin), ranging from between 300 and 1000 km (Figure 3b). The upper limit matches the maximum distance from the margin at the PGM, resulting in no cold-based ice, while the lower limit stops cold-based ice forming at the margin within the range of likely ice streaming. Secondly, we control the shear stress value of the region with a parameter ranging between 120 and 200 kPa.

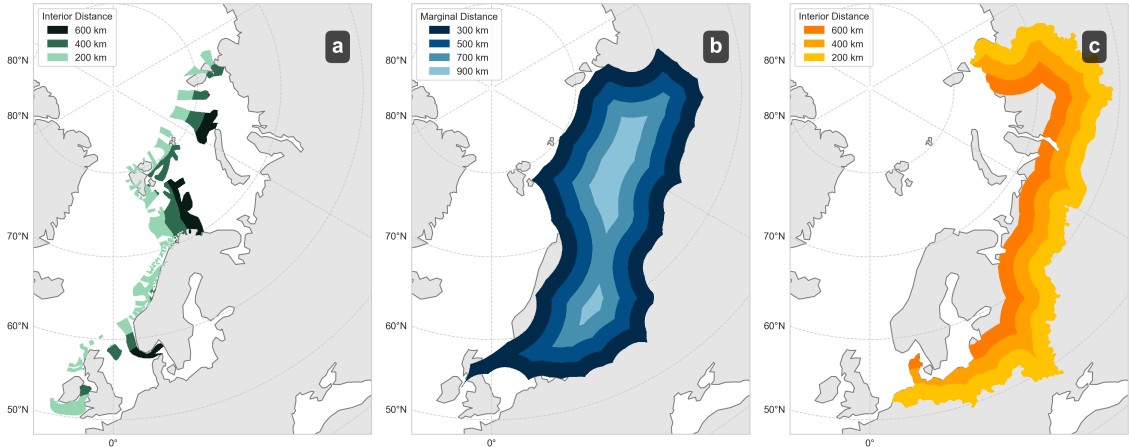

**Figure 3.** Shear stress basal modifications demonstrated using the PGM map and margin: (a) Map of active PGM ice streams for different values of the interior distance parameter. (b) Map of PGM cold-based region for a range of marginal distances, which introduces a frozen bed sector at the interior of an ice sheet. (c) Map of PGM hybrid ice streaming region for a range of interior distances that approximates ice streaming at the southern margin of the ice sheet.

Ice streaming occurred at different times and locations through the Last Deglaciation of the Eurasian ice sheet (Figure 2b). Ice streaming is also likely to occur at periods during ice-sheet advance or retreat of Preceding Glacial (Lang et al., 2018), but a limited amount of geomorphological evidence means it is much harder to constrain when and where. As discussed above, it is not sufficient to simply use the LGM ice stream locations, as these ice streams may not reach the PGM margin and it is not realistic to have an ice stream that terminates within the ice sheet. We therefore introduce an 'active ice streaming distance' parameter for the northern portion of the ice sheet to restrict ice streaming to within a particular distance of the margin ranging between 0 and 1000 km, based upon work by Margold et al. (2015), and so induce a marginal dependence on the previously static shear stress input (Figure 3a). We also introduce a hybrid ice streaming modification to represent the shear stress values that would result from streaming at the southern margin without exact prescription of stream locations. We define a distance from the margin that represent the average length of an ice stream ranging between 0 and 600 km (Margold et al., 2015; Stokes and Clark, 1999) (Figure 3c), and also prescribe a shear stress range whose minimum and maximum values are dependent on the shear stress values for ice streaming and onshore sediment respectively, acting as a proxy for ice-stream density.

In addition to better capture the resulting shear-stress implications of basal driven interactions, the introduction of these three basal modifications allow us to expand the range of ice-sheet geometries and volumes that can be produced by ICESHEET for a given margin; and improve the physical plausibility of the shear stress input by increasing the dimension of our parameter space improving our ability to calibrate the model output and widen the range of uncertainty that can be considered. In total, we describe our shear stress input through 9 parameters.

**Table 1.** Parameters controlling the model inputs for ICESHEET with ranges of values sampled in our ensemble of simulations.

| Parameter Name | Value | Unit | Model Input | Time Period |
|---|---|---|---|---|
| Margin Extent | 0.0 – 1.0 | 1 | Margin | PGM |
| Topographic Deformation | 0.475 - 1.0 | 1 | Topography | PGM |
| Marine Sediment Shear Stress | 10 – 30 | kPa | Shear Stress | PGM and LD |
| Onshore Sediment Shear Stress | 30 – 100 | kPa | Shear Stress | PGM and LD |
| Bedrock Shear Stress | 100 – 150 | kPa | Shear Stress | PGM and LD |
| Ice Streaming Shear Stress | 5 – 20 | kPa | Shear Stress | PGM and LD |
| Ice Streaming Interior Distance | 0 – 1000 | km | Shear Stress | PGM and LD |
| Cold Based Ice Shear Stress | 120 – 200 | kPa | Shear Stress | PGM and LD |
| Cold Based Ice Marginal Distance | 300 – 1000 | km | Shear Stress | PGM |
| Hybrid Ice Streaming Shear Stress | 5 – 100 | kPa | Shear Stress | PGM |
| Hybrid Ice Streaming Marginal Distance | 0 – 600 | km | Shear Stress | PGM |

## 4 Last Deglaciation Reconstruction and Calibration

### 4.1 Ensemble Design

We employ a random Latin Hypercube Sampling (LHS) design to select a 200-member set of input parameter values from the 7-dimensional parameter space controlling the shear stress input (Table 1), after excluding hybrid ice streaming shear stress and marginal distance parameters since the position of southern margin ice streams are prescribed for the LGM (Figure 2b). LHS is a design method, common in Bayesian uncertainty quantification, that efficiently explores the input parameter space to construct ensembles of model simulations (Gregoire et al., 2016; Williamson et al., 2013, 2015). It is typical to sample a minimum of 10x the number of parameters, but a higher sample density is beneficial, particularly if parameter ranges are wide and poorly constrained, hence our large sample design. For each reconstruction and time period, this parameter set is used in combination with the extracted ice margin to generate a corresponding shear stress map. We run 200 simulations for each reconstruction and each of the 4 selected time periods (22, 20, 18, 16 ka), totalling 1600 simulations (Figure 5 and Figure A1).

### 4.2 Calculating Implausibility

GIA models are sensitive to regional distributions of ice-mass loading more so than localised differences in the ice-sheet profile. Since our work is aimed towards developing a GIA ice-sheet input, we choose to assess and calibrate ICESHEET against the ice-sheet volume integrated over three ice-sheet regions which allows us to assess volume difference at a regional scale, rather than over the whole ice sheet or cell-by-cell: Barents-Kara Sea, British-Irish, and Fennoscandia (Figure A1). To assess the model simulations against ICE-6G and GLAC-1D, we use an implausibility metric routinely used in history matching (Williamson et al., 2013). The implausibility is akin to a root mean squared error normalised by a measure of

acceptable discrepancy between a given observation $z$ and modelled value $\mathcal{F}(\hat{p})$, where $\mathcal{F}$ is the model and $\hat{p}$ is a set of model parameters, for a quantity of interest (e.g ice volume) given the known uncertainty in the observation and model limitations. The difference between an observation $z$ and the real system $y$ is quantified by the observational error $e$, such that $z = y + e$, while the difference between the modelled value at the theoretical best set of input parameters $\hat{p}^*$ and $y$ is quantified as the structural model discrepancy $\epsilon$, such that $\mathcal{F}(\hat{p}^*) + \epsilon = y$ (Vernon et al., 2022; Bower et al., 2010; Williamson et al., 2017). Additionally, it is often necessary to be able to predict values of $\mathcal{F}(\hat{p})$ by training an emulator $f(\hat{p})$, such that $\mathcal{F}(\hat{p}) = f(\hat{p}) + \omega(\hat{p})$, where $\omega(\hat{p})$ is the emulation uncertainty, to facilitate denser sampling of the model parameter space than is feasible through direct model runs. Here, we emulate multiple quantities $i$ corresponding to volumes of the Eurasian ice sheet over each of the three regions for each time step and margin series. For each quantity $i$, the implausibility $I_i$ of the model for a given parameter combination $\hat{p}$ is expressed as,

$$I_i(\hat{p}) = \sqrt{\frac{(E(f_i(\hat{p})) - E(\epsilon_i) - z_i)^2}{F(Var(e_i) + Var(\epsilon_i)) + Var(\omega(\hat{p}))}} \tag{1}$$

where $E$ is the expectation (i.e. mean), $Var$ is the variance, and $F$ is a scaling factor for the model and observational uncertainties (see explanation below).

In lay terms, the implausibility represents the discrepancy between the "best guess" (i.e. expectation) of the model emulator $E(f_i(\hat{p}))$ and the observation $z_i$, accounting for systematic model bias (represented by the term $E(\epsilon_i)$) and scaled by the sum of the uncertainties in the observation, model and emulator. Thus, implausibility is large if the discrepancy between model and observation is large relative to the uncertainties.

As explained at the start of this section, the quantities of interest that we emulate and calculate implausibility for are the ice-sheet volumes at each simulated time, region, and margin series (GLAC-1D and ICE-6G), resulting in 24 quantities. We derive the "observed" regional ice volumes $z_i$ for each of these quantities from the ICE-6G and GLAC-1D reconstructions; obtain each set of $\mathcal{F}_i(\hat{p})$ from the corresponding ICESHEET model ensemble; and train a Gaussian Process emulator $f_i$ for each quantity, resulting in 24 emulators of ice volume.

$E(f_i(\hat{p}))$ and $Var(\omega(\hat{p}))$ are calculated as the mean and variance from the emulated volumes $f_i(\hat{p})$, where $Var(\omega(\hat{p})) = 0$ and $E(f_i(\hat{p})) = \mathcal{F}_i(\hat{p})$ for values of input parameters $\hat{p}$ run in the original ICESHEET ensembles. The model bias $E(\epsilon_i)$ and structural uncertainty $Var(\epsilon_i)$ are estimated as the mean and variance of the residuals from the 20 ICESHEET ensemble members with the lowest RMSE against the corresponding GLAC-1D and ICE-6G thickness fields. Since we only have two target reconstructions of ice volume from GLAC-1D and ICE-6G, we choose to estimate $Var(e_i)$ as half the difference between the GLAC-1D and ICE-6G volumes for a given region and time, knowing that this quantity underestimates the true uncertainty in the observations. We therefore choose to augment the observation and model structural uncertainties by 20%, by setting $F = 1.2$. The choice of regional ice-sheet volumes as our metrics, the selection of the $F$ value, and judgement of their impacts of parameter space refinement, is an iterative process and other applications may choose different metrics or tolerance for model discrepancy.

Following from Equation 1, we combine our implausibility metrics into a single implausibility $I(\hat{p})$ for a given set of input parameters $\hat{p}$ such that,

$$I(\hat{p}) = \frac{1}{N_i} \sum_i I_i(\hat{p}) \tag{2}$$

where $N_i = 24$ is the total number of implausibility metrics. In other words, the overall implausibility is set as the mean of the implausibilities calculated for each time, region and margin.

$I(\hat{p})$ is therefore an average measure of how well a particular set of input parameters is able to produce an output via ICESHEET that matches our expectation of ice-sheet volume for each regions, time, and margin considered. We restrict our NROY space to parameter values that correspond to models runs with implausibility $I(\hat{p})$ less than 3, following the Pukelsheim (2012) three-sigma rule typically used in Bayesian History Matching (Andrianakis et al., 2015; Williamson et al., 2015).

### 4.3   Results

GLAC-1D and ICE-6G reconstructions have volume estimates of comparable magnitudes for each time considered, with ICE-6G having volume 105.0%, 97.0%, 112.5% and 115.3% of that of GLAC-1D for 22, 20, 18, and 16 ka respectively. However, the extent of ICE-6G is larger with area 120.4%, 118.6%, 133.4% and 143.2% of that of GLAC-1D for 22, 20, 18, and 16 ka respectively. It appears that producing the smaller ICE-6G area-to-volume ratio is challenging for ICESHEET when used with our shear stress map. This means that, prior to correcting for an esimate of model bias, nearly all ICESHEET$_{6G}$ ensemble

members overestimate the volume of ICE-6G margins, whereas the ICESHEET$_{1D}$ distributions commonly encompass the target GLAC-1D volume. Overall, for most regions and times, the reconstruction target ice volume falls within the distributions of modelled volumes, often towards the lower values. There is a significant lack of overlap between the ICE-6G target ice-sheet volume and the reconstructed volume using ICE-6G margins in the British-Irish sector. This model-data discrepancy is accounted for in the prescription of the model bias correction (Equation 1) which reduces the influence of this misfit on

the overall implausibility metric. The algorithm used to extract ice-sheet margins from the target reconstructions leads to some differences in extent, such as an overestimation in the Barents Sea extent at 16 ka (Figure 5i,j). This is as a result of the smoothing procedure applied during margin creation which can lead to underestimation where there are small thickness protrusions, and overestimation at some concave margin edges.

      Before applying our criteria for implausibility, we find that the 200-member ensemble generated for the Last Deglaciation

has a mean implausibility of $4 \pm 2$. After removing members with implausibility of greater than 3, we find that 116 members have been excluded, leaving 42% of parameter points within the NROY space, and a new mean implausibility of $2.1 \pm 0.4$. The NROY space favours reduced ice-sheet volumes with all times considered for ICESHEET$_{1D}$ and ICESHEET$_{6G}$ exhibiting a reduction in average total volume compared with the original distributions (Figure 4). In this work, we express ice-sheet volumes in terms of sea-level equivalent (SLE) volume which we calculate by dividing a given ice-sheet volume by modern

ocean area. We find that the largest mean percentage reduction in volume is for the Fennoscandian region of the ICESHEET$_{6G}$ 18 ka margin at $-16\%$, while the least reduced is the British-Irish region of the ICESHEET$_{1D}$ 16 ka margin at $-4\%$. The

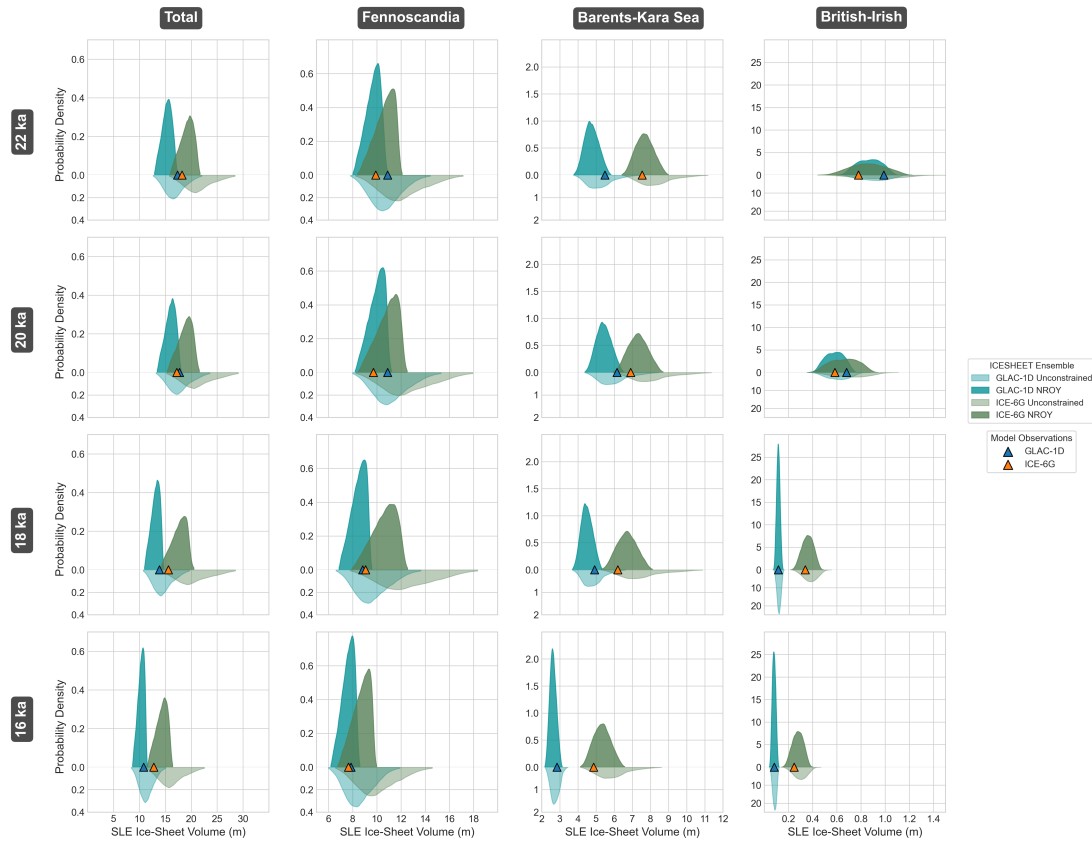

**Figure 4.** Deglacial ice-sheet volume probability density functions, derived from a $10^5$ member sample from the quantity's associated Gaussian Process emulator, for each region (total, Barents-kara Sea, Fennoscandia and British-Irish) and time (16, 18, 20, and 22 ka) after correcting for model bias. ICESHEET$_{1D}$ (blue) and ICESHEET$_{6G}$ (green) are shown separately for both before (lighter shade, below) and after (dark shade, above) applying the history matching NROY parameter constraint. The blue and orange triangle symbols show the target regional ice volumes from the ICE-6G and GLAC-1D reconstructions respectively.

maximum volume across all margins and times is reduced from 29.7 m to 21.1 m after history matching, with the minimum increased slightly form $8.7$ from $8.9$ m SLE (Figure 4).

Prior to applying the bias correction fields, ice-sheet thickness in the interior of the Barents-Kara Sea region is consistently underestimated over all margins and times, potentially due to the lack of modelled dynamic that are important for marine ice sheets, but shows lower variance than other regions (Figure A4). The largest variance occurs in ice-sheet thickness the centre of the Fennoscandian region. However, thickness in this region appears to be over-estimated in ICE-6G and underestimated in GLAC-1D simulations. In addition, both target reconstructions position ice-sheet domes slightly towards the marine margins and exhibit thinner continental marginal ice. This likely reflects the larger accumulation of snow closer to the

coast and the influence of a rain shadow in reducing accumulation towards the interior. In contrast, since ICESHEET doesn't

see the effect of climate on the ice-sheet geometry, our simulated position of the ice dome is very central, yet this discrepancy is consistent between model and reconstructions and of a similar order of magnitude to the discrepancy between the two target reconstructions. The ice thickness at the margin is systematically thicker in our simulations than in both reconstructions. Because of our choice of metric, history matching against regional volume, we therefore preference ice sheets that are thinner in the interior and thicker at the edges but a different target metric would preference simulations differently, such as max thickness which would likely select thinner overall simulations. Regional differences also exist in post-history matching mean model performance after removal of the model bias field. GLAC-1D and ICE-6G remain respectively under and overestimated, while the primary misfit is now in the centre of the Fennoscandian ice-sheet, likely due to the large disagreement between GLAC-1D and ICE-6G in this region.

To better understand the relationship between implausibility and the shear-stress input parameter values, we generate an optical depth image which reveals the shape of the NROY region within our parameter space (Figure 6). This image shows the density of NROY parameter values, and the minimum implausibility, across each of the 21 faces of the 7-dimensional parameter hypercube. Each face is associated with a parameter pair and consists of 1600 (40x40) pixels. For a given face, each pixel represents 2 fixed values for the 2 parameters associated with the face, and the pixel's NROY density and minimum implausibility values are derived from a 1000-member random sample of the 5 remaining unfixed parameters. Each 1000-member sample is evaluated using the 24 Gaussian Process emulators in order to calculate their associated implausibility values, meaning that each face in the 21 face image is the result of 38.4 million emulator evaluations.

Our analysis reveals that there is a slight preference for lower ice stream and marine shear stress values and a relatively strong preference for onshore shear stress values (Figure 6). This is likely due to the smaller ice-sheet geometries that these lower values result in. Bedrock shear stress values show no clear relationship indicating insensitivity of our regional ice-sheet volume metrics to this parameter. This may be due to the small relative area covered by bedrock, in contrast to other types of shear stress categories, resulting in a limited impact on ice-sheet volume.

We see a strong indication that lower values of onshore sediment shear stress (mean of $55.3 \pm 19.6$ kPa), and higher values of cold-ice interior distance (mean of $786 \pm 138$ km) are favourable (Figure 6). A large value of the cold-ice interior distance parameter will produce a smaller area of cold-based ice, since this distance is defined from the margin inwards, and so such runs will produce smaller ice-sheet volumes that have lower implausibility. In addition, lower overall shear stress values are shown to be more realistic in most cases but we do not see the same relationship with cold-ice shear stress and bedrock, as this model seems insensitive to these parameters. Finally, a preference for higher values of ice streaming interior distance (mean of $627 \pm 244$ km), indicates that longer ice streams, and therefore thinner ice, is preferred. We find that these parameter distributions are common throughout the deglaciation, but with a stronger influence of cold-ice interior distance for smaller ice sheets in the later deglaciation stages. We hypothesise that is a result of the ice sheet being in a state of climate disequilibrium in the later stages of the deglaciation which may have caused a thick yet narrow ice-sheet geometry due to ice melt at the margins while the thick interior is still present. We also observe a larger influence of marine sediment shear stress on models with a greater margin extent because smaller extents have less ice that covers marine sediments.

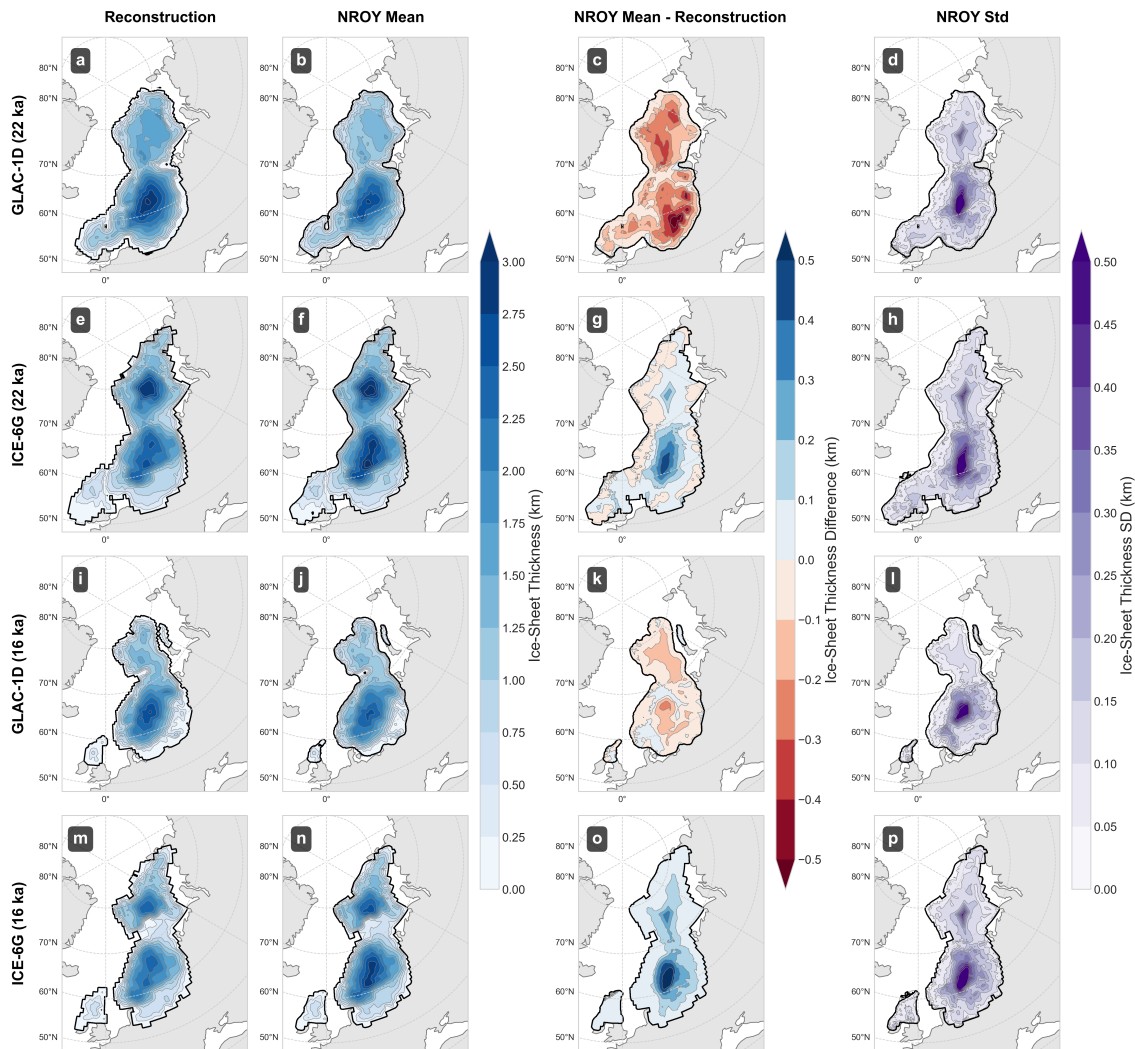

**Figure 5.** Comparison of the constrained NROY ensemble of ICESHEET$_{1D}$ and ICESHEET$_{6G}$ simulations, with model bias removed, against the GLAC-1D (first row: a-d) and ICE-6G (second row: e-h) reconstructions respectively, for the 22 ka time slice. (a) GLAC-1D target reconstruction. (b) Mean of the NROY ensemble of ICESHEET model outputs, with model bias removed, using the margin derived from a. (c) Difference between our ensemble mean (b) and the target reconstruction (a). (d) Standard deviation of this ensemble. Panels e-h are as a-d but for the 16 ka time slice.

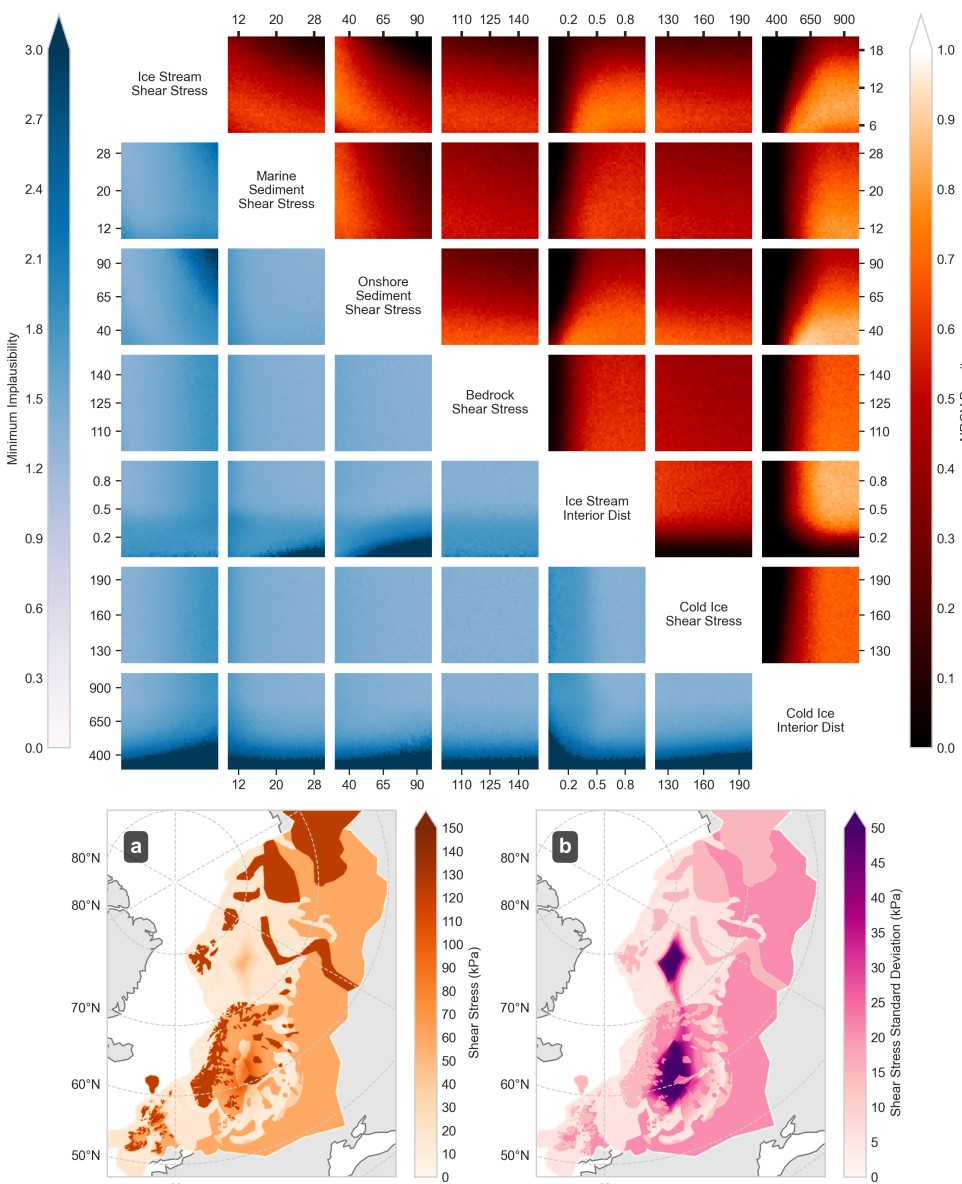

**Figure 6.** Shape of the NROY parameter space (Sect. 4.2) for the Last Deglaciation. Density of the NROY space (reds) and the minimum implausibility value (blues) shown for each face of the 7-dimensional hypercube. Each panel is composed of 40x40 pixels, while the value at each cell is derived from a 1000-member random sample of 24 Gaussian Process volume emulators (1 for each time, margin, and region) in order to calculate the resulting implausibility and derive a value for each pixel. Maps show the resulting mean (a) and standard deviation (b) of the NROY shear stress map input averaged over ICESHEET$_{6G}$ and ICESHEET$_{1D}$ at 22 ka.

## 5 Application to the Penultimate Glacial Maximum

### 5.1 Initial Model of the Penultimate Glacial Maximum Eurasian Ice Sheet

In order to model the configuration of the PGM Eurasian ice sheet, and to include new parameters controlling hybrid ice streaming, marginal extent, and topographic deformation, we first generate a new 1000-member, uniform LHS sample of the model parameter ranges as detailed in Table 1. Our initial ensemble iteration of PGM ice-sheet simulations is run using modern day topography (Schaffer et al., 2016) (initially ignoring the topographic deformation parameter), and the 1000-member set of generated shear stress map and ice-sheet margin inputs, based on work by Batchelor et al. (2019) (Figure 1e), as described in Sect. 2.4.2. While the margin extent parameter was initially sampled as uniform, in order to aid in training of the Gaussian process emulator, the following volume estimates are reported from an emulation derived sample of $10^5$ parameters using a normal range for the margin extent parameter, centred on 0.5 with a standard deviation of 0.1.

Over the full ensemble, this produces an ice sheet with a volume of $45 \pm 15$ m SLE (Figure 7a) which falls below the $\approx 70$ m SLE value by Colleoni et al. (2016) and within range of the 52.5 m SLE value of Lambeck et al. (2006) and the 33.2 m SLE of de Boer et al. (2013), within uncertainty. Next, we apply corrections for glacial isostatic adjusted topography to the ensemble, and utilise our Last Deglaciation history matching (Sect. 4) to refine our PGM ice-sheet reconstruction.

### 5.2 Effects of Glacial Isostatic Adjustment

Previous research has shown the importance of accounting for GIA when simulating ice sheets with the ICESHEET model (Gowan, 2014) and so we must account for this in our simulations, using the approach outlined in Sect. 2.4.2. We find that, after applying the simple deformation model, scaling the magnitude of deformation by the topography deformation parameter, our mean deformed topography is depressed by a total volume of $4 \pm 1 \times 10^6$ km$^3$ compared to modern day topography which, if this space were filled with ice, would be equivalent to $9 \pm 4$ m SLE. On average, the region covered by ice is depressed by $0.1 \pm 0.2$ km compared with modern day, with areas close to the interior of the ice sheet experiencing the highest levels of deformation, with a maximum depression of $1.2 \pm 0.2$ km (Figure A2). All topography underneath the ice-sheet mass is depressed by applying ELRA but variation in this depression is minimal at the exterior regions of the ice since the model is less sensitive to the smaller changes in ice thickness at the peripheries of the ice sheet.

Deformed topography has a non-negligible impact on the distribution of ice volume in our ensemble with mean volume increasing from $45 \pm 15$ m SLE to $50 \pm 16$ m SLE (Figure 7). A single iteration of the ELRA topography, combined with the deformation scaling parameter, allows us to account for the first order effects of GIA, with our experiments finding that subsequent iterations produce ice volume changes of an order of magnitude less than the first.

### 5.3 Reconstruction of the Penultimate Glacial Maximum Eurasian Ice Sheet

Since the model parameter space is expanded to include parameter controlling hybrid ice-streaming, we first perform a new 1000-member LHS sample for the PGM and simulate these with ICESHEET and ELRA deformed topography. As in Sect.

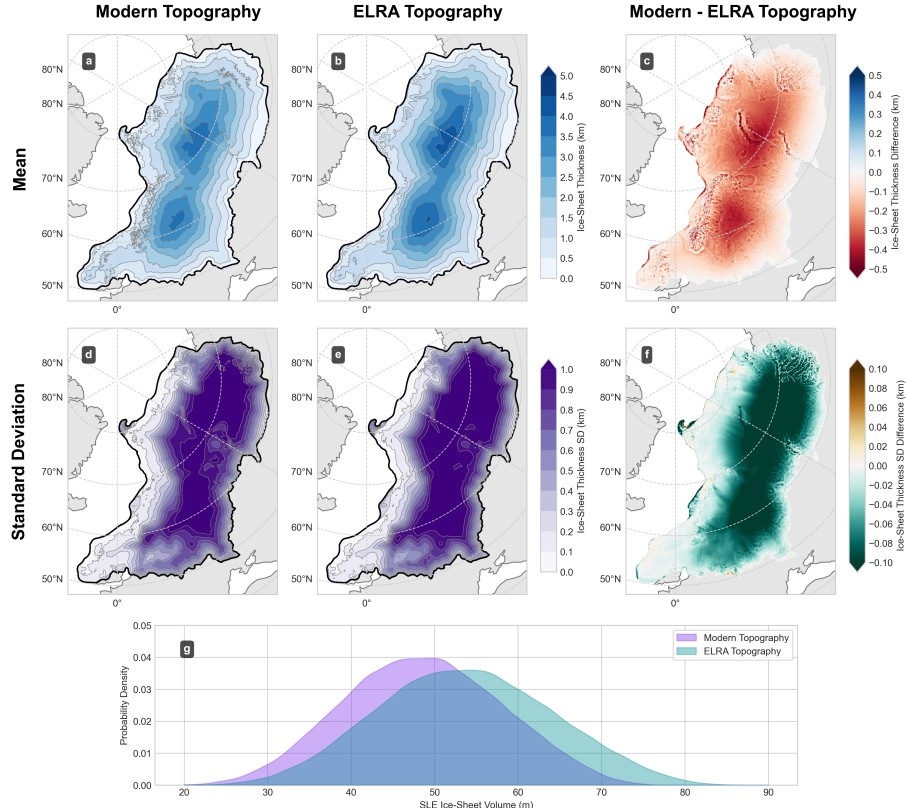

**Figure 7.** Impact of ELRA topography deformation on PGM ice thickness. Mean thickness across the full ensemble using modern day topography (a) and ELRA deformation respectively (b). (c) Difference in the ensemble mean thickness between ELRA adjusted and modern topography simulations. (d) and (f) as a-c but showing ensemble standard deviation. (g) Distributions of PGM ice sheet volume in the ensemble run with modern topography (purple) and ELRA adjusted topography (green).

4.2, we compile our best estimate reconstruction of the PGM with quantified uncertainty by excluding members that have an implausibility of greater than 3. The implausibility values for PGM sample members are derived by utilising the 24 Gaussian Process emulators trained on each volume metric, as in Sect. 4.2, for the 7 common parameters. We account for the presence of a bias term in our initial implausibility by subtracting a scalar bias of 1.76 m from all PGM volumes. This bias was calculated as a result of scaling the NROY PGM volume mean by the mean Last Deglaciation percentage bias at 20 ka.

Applying the NROY constraint acts to reduce the mean of our ice-sheet thickness ensemble from $2.0 \pm 0.4$ km SLE to $1.8 \pm 0.3$ km SLE (Figure 9). Much of this reduction in volume is from favouring ice sheets with a thinner interior (Figure 8). The pre-history matched mean maximum thickness of $4.8 \pm 1.0$ km, occurring in the interior, reduces to $4.3 \pm 0.9$ km, but with slightly thicker ice at the southern margin, compared with the maximum ice thickness over North America at the LGM of 3.38 km, and present-day Greenland and Antarctica at 3.14 km and 4.01 km respectively (Tarasov et al., 2012). After history matching, we see the highest variation in thickness in the NROY subset is in the central eastern portion of the ice sheet, except

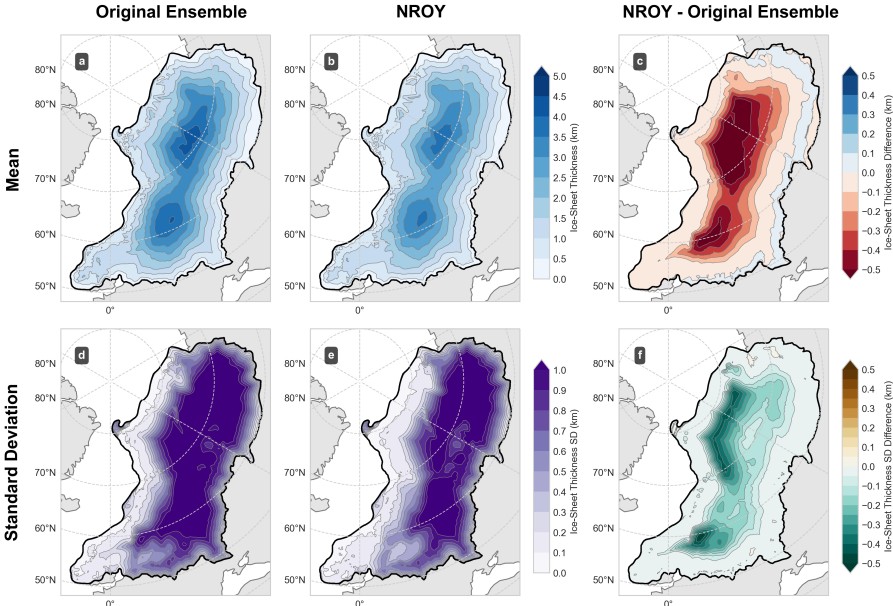

**Figure 8.** Ensemble mean ice-sheet thickness before and after history matching (a) full ensemble and (b) NROY subset, and the difference between these means (c). d-f are as a-c but for the standard deviation instead of mean. Applying constraints on the Last Deglaciation leads to ice sheets with smaller volumes in the ice interior, but slightly thicker ice at the margins.

for the Barents Sea region where cold-based ice is present through many of the accepted ensemble members (Figure 8). In addition, we find that history matching favours a reduction in the shear stress value for the interior of the ice sheet, but an increase in the Siberian sector, while the exterior shear stress values remain similar (Figure A3). Our mean PGM regional ice-
sheet volume is $24 \pm 8$ m SLE for the Barents-Kara Sea ($27 \pm 9$ m SLE pre-history matching), $19 \pm 6$ m SLE for Fennoscandia ($21 \pm 7$ m SLE pre-history matching, and $1.7 \pm 0.2$ m SLE for the British-Irish region ($1.8 \pm 0.2$ m SLE pre-history matching. We find the 5th and 95th percentile of our NROY ice-sheet volume distribution for the PGM to be $35$ m SLE and $62$ m SLE respectively. Our lower value is comparable with the Eurasian ice volume simulated with dynamic ice-sheet modelling by de Boer et al. (2013) of $33.2$ m SLE, and our peak probability ($48$ m SLE) close to the reconstruction by Lambeck et al.
(2006) ($52.5$ m SLE) using GIA inversion methods. The dynamic ice-sheet model output which results in a $70$ m SLE PGM Eurasian ice sheet by Colleoni (2009) exceeds our maximum. Similarly, the simulation developed by Abe-Ouchi et al. (2013), and subsequently used in the PMIP protocol (Menviel et al., 2019), is within the 99th, but greater than our 95th percentile ($\approx 64$ m SLE).

## 6   Discussion

ICESHEET (Gowan et al., 2016a) is able to produce simple, perfectly plastic, steady-state ice-sheet reconstructions with minimal number of inputs. Such reconstructions are appropriate inputs for calculating RSL change since GIA modelling is less

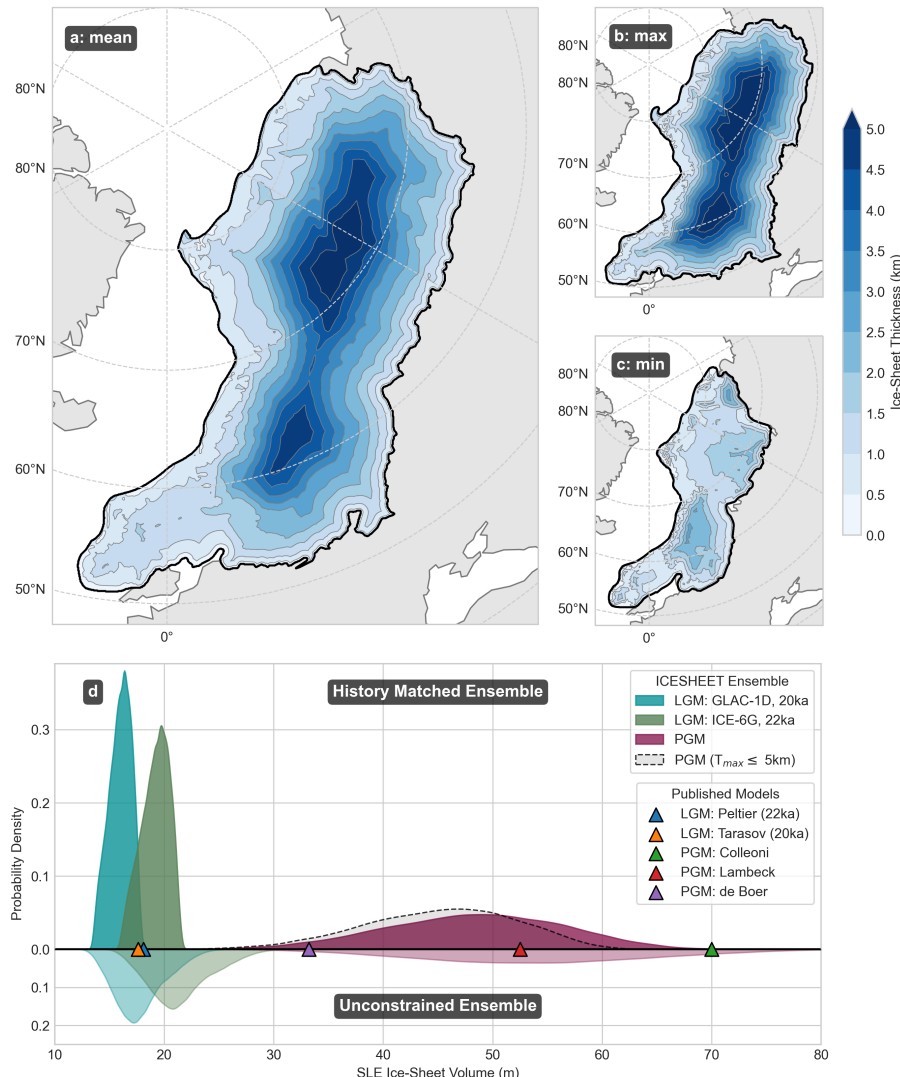

**Figure 9.** (a) Penultimate Glacial Maximum (PGM) Eurasian ice-sheet thickness ensemble member from the constrained (NROY), having been refined using information leveraged from history matching against the Last Deglaciation, ensemble with total ice-sheet volume closest to the probability distribution mean (48 m SLE). Smallest (b) and PGM largest (c) NROY ensemble members after history matching. (d) Probability density functions of unconstrained (bottom, lighter shade) and history matching constrained (top, darker shade) ice-sheet volumes for ensembles of the 20 ka GLAC-1D (blue) and 22 ka ICE-6G Last Glacial Maximum margins and the PGM (purple) compared against published ice-sheet dynamical simulations reconstructions from the corresponding time periods (Colleoni, 2009; Lambeck et al., 2006; de Boer et al., 2013; Peltier et al., 2015; Tarasov et al., 2012). Dashed grey line shows alternative probability density function when we constrain to simulations with ≤ 5 km maximum thickness.

sensitive to the specific surface geometry of an ice sheet and more sensitive to the regional load distribution and evolution. Utilising a history-matching approach and a large ensemble to explore a range of controlling shear stress parameters, we produced an ICESHEET-derived set of simulations for the Last Deglaciation of the Eurasian ice sheet ($17 \pm 2$ m SLE at 22 ka,

averaged across ICE-6G and GLAC-1D margins). These results provide an alternative ice model independent of climate forcing or the need to fit with RSL data, and provide ice-sheet thickness estimates not offered from geomorphologicaly-constrained margin reconstructions (Batchelor et al., 2019; Hughes et al., 2016). These LGM outputs then help to constrain a reconstruction of the PGM Eurasian ice sheet, where constraints on ice-sheet extent, thickness and basal conditions are far more limited. Our final model outputs suggest an ice-sheet volume of $48 \pm 8$ m SLE, which is 2 to 3.5 times larger than that for the Eurasian ice

sheet at the LGM. Between the LGM and PGM simulations, the Barents-Kara Sea region has the highest average percentage increase in volume at $+245\%$ (from $7 \pm 1$ m to $24 \pm 8$ m SLE), followed by the British-Irish region at $+170\%$ (from $0.6 \pm 0.1$ m to $1.7 \pm 0.2$ m SLE) and the Fennoscandia at $+63\%$ (from $11 \pm 2$ m to $19 \pm 6$ m SLE). If we combine our Eurasian ice-sheet reconstruction for the PGM with LGM values of the other ice sheets averaged from ICE-6G (Peltier et al., 2015) and GLAC-1D (Tarasov et al., 2012) (9.5 m, 78.8 m and 72.9 m SLE excess ice volumes of the Greenland, Antarctic and Laurentide ice

sheets, respectively), we arrive at an ice volume that is 7 m SLE higher than the value suggested by the delta $^{18}$O curve for MIS 6 (Waelbroeck et al., 2002). This would suggest that balancing the total ice volume during the PGM would require a $\approx 10\%$ decrease in the size of the Laurentide ice sheet compared to the LGM. This spatial difference in the distribution of ice load between the LGM and PGM across Eurasia and North America has significant implications for the pattern and magnitude of Last Interglacial sea level (Dendy et al., 2017), compared to the Holocene. It should be noted that this simple comparison is

made to illustrate the implications of our results on the relative size of the Laurentide ice sheet, but with the caveat that the relationship between global average sea level and global ice-sheet volume is more complicated than implied here, due to the effects of ocean-load driven bathymetry changes and ice-sheet driven topography changes modifying ocean basin volumes. This mean that estimates of global mean sea level are dependent on assumptions of the visco-elastic response of the Earth, and may infact differ by up to 20 m from the estimate used here (Gowan et al., 2021).

One limitation of our approach is that ICESHEET does not represent dynamic ice-sheet processes or climate information that may be important for defining spatial variations in Eurasian ice geometry at the PGM. In our reconstruction, the location of ice domes remain central relative to the ice-sheet margin, which in turn is prescribed as a maximum synchronous extent and, by extension, volume. In contrast, Colleoni (2009) do include dynamics in their ice-sheet reconstruction but a near implausible total SLE ice-sheet volume of 70 m (since this would require a Laurentide ice sheet $40\%$ smaller than at the LGM which seems

unlikely), combined with large uncertainties on required climate inputs, casts doubt on the reliability of this simulation for use in climate and GIA model inputs. By utilising a range of ice margins (Figure 1) our outputs do considering the potential varying size of the Eurasian ice sheet maximum during the late Saalian (Ehlers et al., 2011), though analysis of the consequence of spatial and temporal variations during the Penultimate Deglaciation on GIA must be considered in future work.

The use of a shear stress map to represent bed friction, decomposed into key parameters, provides a flexible framework

for reconstruction Quaternary Eurasian ice-sheet geometries since the parameter space can be easily and quickly explored to produce large ensembles of simulations that span the uncertainty in this input. Ice-sheet processes at the bed often manifest as a

change in basal shear stress (Knight, 1997) and approximations to the basal implications of such processes can be incorporated into this framework, for example by approximating the presence of cold-based ice. Uncertainty in the location of sediment types, bedrock and ice streaming remains a challenge but we find that use of variable density regions, such as the hybrid ice streaming component employed in the southern sector of the ice sheet, have a strong control on the implausibility metric and can therefore be used to effectively explore the exploring the impact of these uncertainties The shear stress map is an attempt to represent a complex distribution of basal properties (Knight, 1997). Our work has expanded this methodology to include the cold-based ice and active ice streaming basal modifications which have had a strong impact on the implausibility metric, improving the simulation fit during history matching when applied to the Last Deglaciation, with the exception of the British ice sheet (Figure 4) where simulation mismatch is likely due to discrepancies in ice-margin extraction. By extension, this approach also worked to better refine our reconstructions at the PGM. The modelling framework could be further improved by validating these modifications against other ice-sheet models, such as for the Laurentide and Greenland ice sheets.

By employing history matching, leveraging information from models of the Last Deglaciation, we were able to refine the ensemble mean for our PGM ensemble from $50 \pm 16$ m to $48 \pm 8$ m SLE. This approach reduced the size of our original parameter space, which had initially produced a broad range of ice-sheet volumes, by 58% and revealed a tendency for our ensemble to overestimate ice-sheet thickness since our refined ensemble preferred thinner ice sheets. This technique could be improved in a number of ways. The average relative distance in regional volume metrics derived between our two target reconstructions is 15.9%. However, some metrics are much more uncertain, such as the volume of the British-Irish ice sheet at $20$ ka, which has a relative distance of 76.0%. It would be beneficial to extend the model comparisons beyond GLAC-1D and ICE-6G, such as to work by Patton et al. (2017). In addition, the GLAC-1D target reconstruction is itself derived from an ensemble of simulation (Tarasov et al., 2012). Therefore, the observation metric uncertainty could be more accurately accounted for in our procedures if the original ensembles from which the target reconstructions are derived could be obtained .

A possible criticism of our work is that the PGM ice sheet we are predicting with our model is "out of sample" compared to the Last Deglaciation that we have calibrated the model on since the PGM ice sheet is larger than at the LGM. This is a very common situation in modelling uncertainty quantification work. We believe this analysis is robust to this issue since the ice sheet volume is correlated with extent meaning that, since our simulations are based on the same shear stress map and modifications, the history matched parameter space is applicable for simulation of both the Last Deglaciation and the PGM. However, given the larger PGM margin, ICESHEET is able to generate ice-sheet thickness values that may be physically implausible (greater than $5$ km). We investigate the effect that constraining to simulations that have a maximum thickness of $\leq 5$ km has on our PGM volume probability density function (Figure 9) and find that this results in a reduced volume estimate of $45 \pm 7$ m SLE.

This work has demonstrated the benefit of using simpler ice-sheet models with a Bayesian uncertainty quantification framework to explore the range of uncertainty in periods when there are highly uncertain ice-sheet geometries. This workflow, using ICESHEET and history matching, could be applied to other regions (e.g. Laurentide) or times (e.g. the large MIS 12 ice sheets) where there are also large uncertainties in extent, thickness and timing.

## 7 Conclusions

By employing a simple ice-sheet model we were able to investigate the range of physically plausible PGM ice geometries for the Eurasian ice sheet. The primary control on geometry changes are due to the 2D shear stress map input that we decompose into 9 parameters controlling regional shear stress values and the shear stress influence of key basal processes. By employing a Latin Hypercube Sampling technique, we explore the range of possible ice-sheet thickness distributions over this parameter space. We find that our model procedure generates a PGM ice-sheet ensemble with a total SLE volume range of $50 \pm 16$ m SLE. In order to refine this ensemble range, we employ a history matching procedure, leveraging information from previously published reconstructions of the Last Deglaciation, in order to rule out combination of input parameter values that produce unrealistic ice sheets.

This work is aimed at producing ice sheet simulations to be used as input to sea-level models and so therefore assess ice-sheet geometry at a regional scale that ignores local details in the thickness profile. History matching rules out 58% of our parameter space and heavily favoured parameter combinations that lead to smaller ice sheet configurations. We applied the refined parameter space (NROY space) to our original PGM ensemble, reducing the mean and uncertainty on our range of PGM volume to $48 \pm 8$ m SLE. This refinement reflects the preference for smaller Eurasian ice sheets found in the Last Deglaciation history matching procedure and points at the tendency for ICESHEET driven by a parameterised shear stress map to overestimate ice sheet thickness. This work is currently limited to a single synchronous maximum but can be applied to develop reconstructions of ice extent and thickness over a full deglacial cycle that can in turn serve as input into a GIA model for predicting changes in RSL. The rate and timing of the deglaciation are important factors in the pattern and magnitude of RSL change during deglaciation and the subsequent interglacial and, despite the lack of chronological constrains, producing a full Penultimate Deglaciation history for Eurasia remains an important challenge to overcome in future work.

*Code and data availability.* The data associated with this paper are openly available from the University of Leeds Data Repository, https://doi.org/10.5518/1287. Code used to generate the data and figures are available from Zenodo, https://doi.org/10.5281/zenodo.7544619. Modified ICESHEET model used in this work available at https://github.com/oliverpollard/icesheet.

## Appendix A: Figures and Margin Extraction Algorithm

In order to perform a history matching procedure with ICESHEET we require that the ice-sheet margins used as input be approximately equivalent to those of the reconstructions we are comparing to. Margins are not provided explicitly with either the ICE-6G or GLAC-1D reconstructions and so we instead developed a simple algorithm to extract margins from gridded ice-sheet thickness rasters. The procedure is as follows:

For each reconstruction and time period, we first reproject and interpolate the ice-thickness and topography fields from their native grid to our LAEA model grid using bilinear interpolation. We then extract the ice margin from the gridded ice thickness field using an algorithm that first identifies grid cells at the edge of the ice sheet from the ice thickness field, then employs

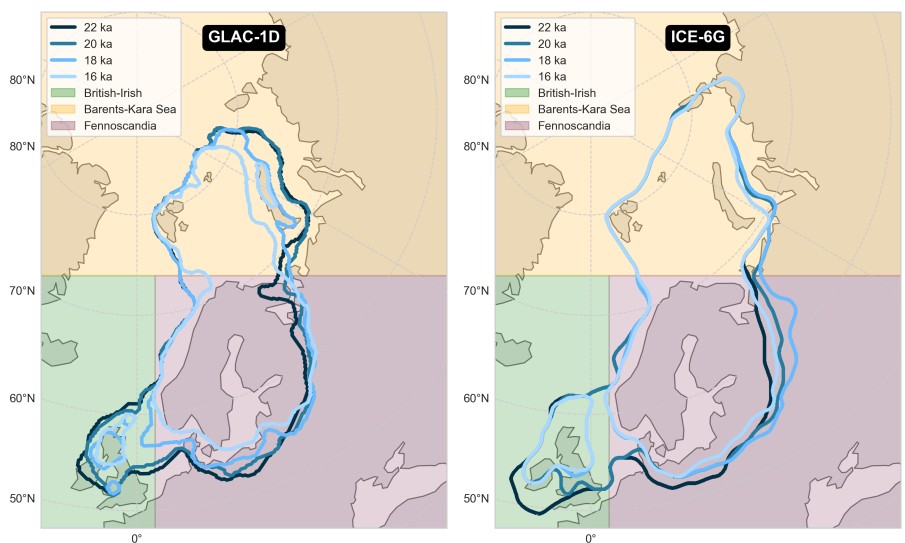

**Figure A1.** Left figure shows GLAC-1D margins for 22, 20, 18 and 16 ka as extracted by the algorithm described in Sec. A. Right figure as left but for ICE-6G. British-Irish (green), Barents-Kara Sea (yellow) and Fennoscandia (red) region divisions are shown.

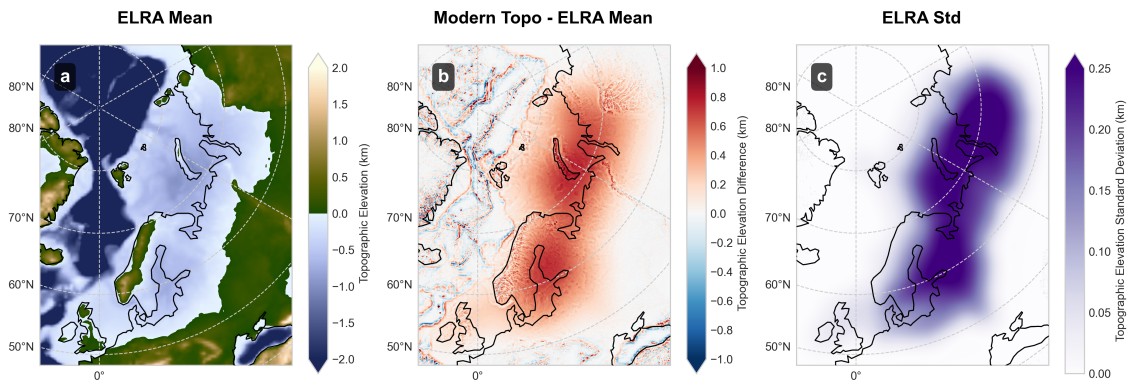

**Figure A2.** Topography Deformation. Mean (a) and standard deviation (b) of topography across the full ice sheet ensemble and (c) difference between the ensemble mean and modern day topography.

a pathfinding procedure to order the collected cells into an ordered polygon structure, and finally converts the ordered cell positions into coordinates. In addition a region mask, minimum considered thickness value, average ice-sheet thickness value, and a median filter smoothing may be applied as conditions.

1. The 2D ice thickness is converted into a binary image (or mask), with values of 1 where ice is present, and 0 where it is not, using a minimum thickness value defined as a parameter.

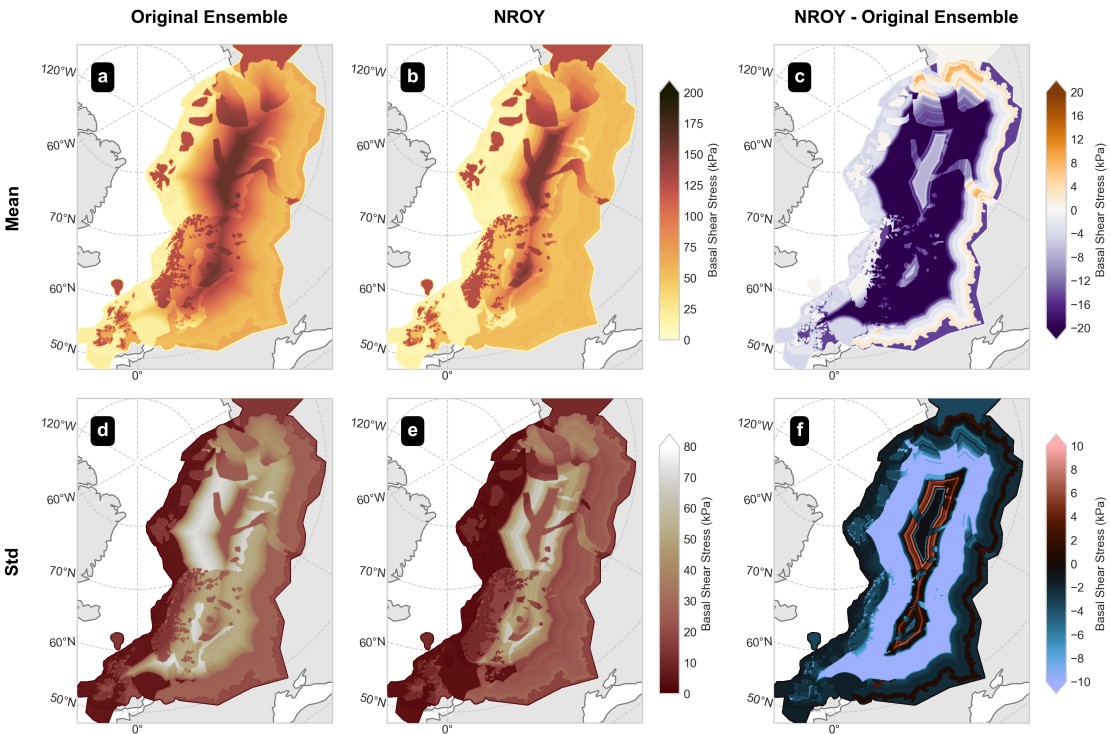

**Figure A3.** Ensemble mean basal shear stress before (a) and after (b) history matching and (c) difference between these means. Panels d-f are as a-c but for standard deviation.

2. The binary image may optionally filtered by another mask, such as a mask defining the continental shelf, to restrict the area of the margin.

3. We perform a binary erosion morphology operation (REF) on the binary image, using a structuring element with square connectivity equal to 1, to reveal the binary shape of the ice that is 1 grid-cell smaller than the original.

4. The binary-eroded image is subtracted from the original binary image to reveal a binary outline of the ice-sheet margin.

5. Each margin cell is then checked via a recursive procedure to identify those cells adjacent to it that form part of a continuous path. The set and order of cells that form each path are then stored. Once assigned to a path, a cell is not considered by the algorithm for future paths.

6. The set of ordered cell paths are then converted, in combination with their cell coordinates, to polygon geometry objects.

7. Each polygon may be optionally checked for the average ice-thickness it contains value, specified as a parameter, in order to exclude patches of thin, disconnected ice.

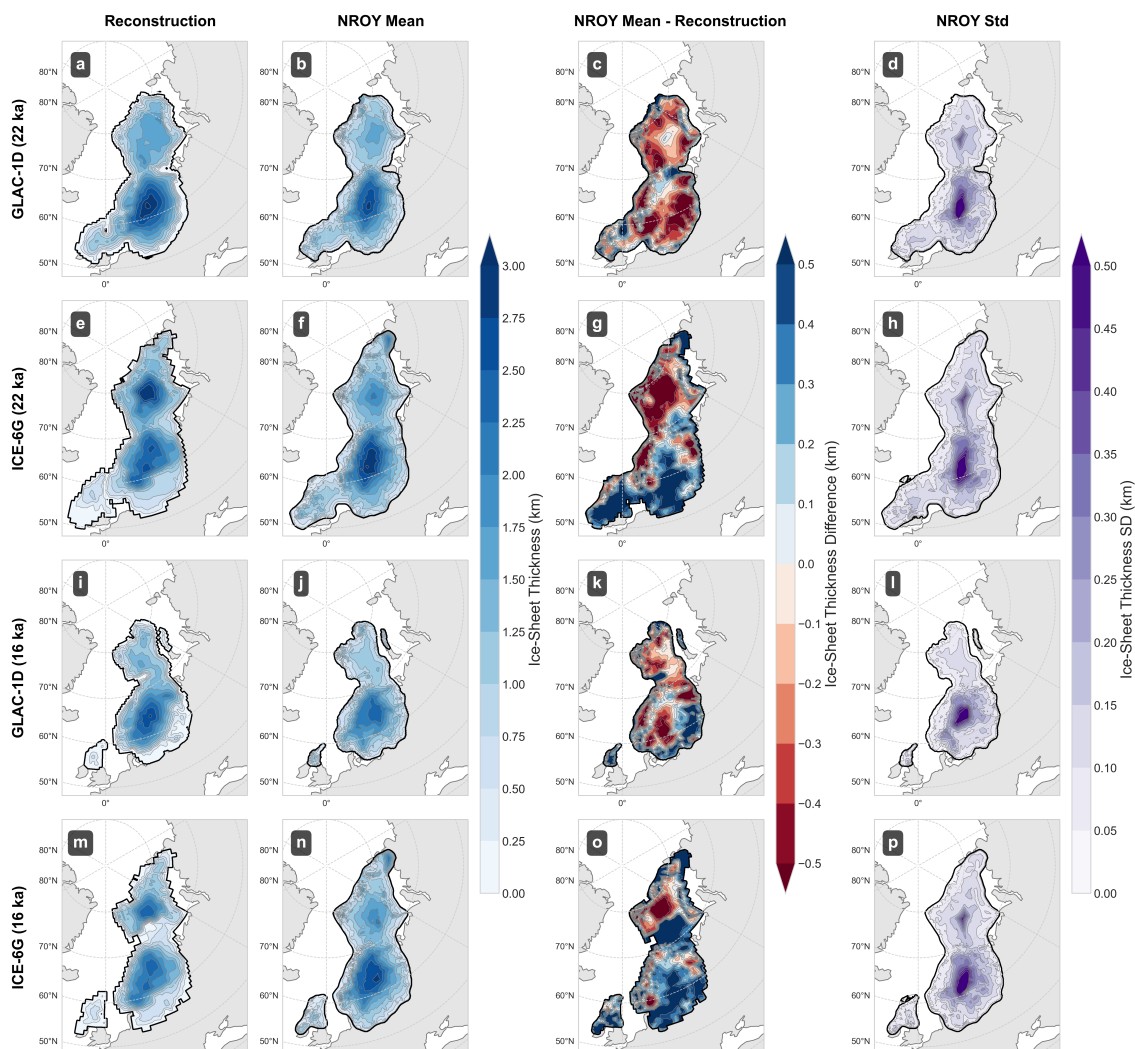

**Figure A4.** Comparison of the constrained NROY ensemble of ICESHEET$_{1D}$ and ICESHEET$_{6G}$ simulations, before removal of the model bias, against the GLAC-1D (first row: a-d) and ICE-6G (second row: e-h) reconstructions respectively, for the 22 ka time slice. (a) GLAC-1D target reconstruction. (b) Mean of the NROY ensemble of ICESHEET model outputs using the margin derived from a. (c) Difference between our ensemble mean (b) and the target reconstruction (a). (d) Standard deviation of this ensemble. Panels e-h are as a-d but for the 16 ka time slice.

8. If an optional smoothing value is specified, an iterative smoothing procedure is performed whereby the newly calculated margin polygons are regridded onto a fine grid which is then smoothed with a median filter of size specified by the smoothing value, and then reperforms steps 1-7 to calculate a smoothed set of margin contours.

*Author contributions.*  OGP produced the computational experimental design (alongside NLMB, LJG, NG, and LCA); wrote and executed the code required for preparing the experiments, creating modified shear stress maps, extracting and generating ice-sheet margins, output processing and analytics, and filtering parameter spaces; modified and ran the ICESHEET model, and trained Gaussian processes emulators; visualised and analysed the results and wrote the manuscript with contributions from all other authors (including major contributions from NLMB and LJG). NLMB acquired the funding, was the source of the idea behind the work, supervised the work throughout, provided expertise in palaeo datasets, and made major contributions to the interpretation of results. LJG co-supervised and provided expertise in numerical experimental design, Bayesian uncertainty quantification, ideas for many of the computational solutions developed, majorly contributed to interpretation of results and the subsequent direction of experimental investigation. NG co-supervised and provided expertise in glacial isostatic adjustment, contributed to experimental design decision, and made particular contributions to the treatment of topography. VC expanded the Last Deglaciation Eurasian shear stress map to encompass the region covered by the Penultimate Glacial Maximum Eurasian ice sheet, provided expertise on geomorphological features in the southern North Sea as well as in the design of shear stress modifications. JCE provided the Last Deglaciation Eurasian shear stress map, shear stress value ranges, and contributed expertise surrounding shear stress, ice-sheet model operation, and experimental design. LCA provided expertise in Bayesian uncertainty quantification, large ensemble experiments, and implausibility metrics.

*Competing interests.*  The authors declare that they have have no conflict of interest.

*Acknowledgements.*  This paper forms a contribution to the RISeR project, a European Research Council (ERC) project funded under the European Union's Horizon 2020 research and innovation programme (grant agreement no. 802281), supporting N.B. O.P. and V.C.. The authors acknowledge PALSEA, a working group of the International Union for Quaternary Sciences (INQUA) and Past Global Changes (PAGES), which in turn received support from the Swiss Academy of Sciences and the Chinese Academy of Sciences. The authors wish to thank Sarah Bradley, Amy McGuire, Graham Rush and the University of Leeds Climate-Ice Group for fruitful discussion around this work, and Evan Gowan, Lev Tarasov and Samuel Toucanne for their comments that helped improved the manuscript. The Leeds Centre for Environmental Modelling And Computation (CEMAC) provided invaluable technical assistance for this project. L.J.G. is funded by a UK Research and Innovation Future Leaders Fellowship (MR/S016961/1). N.G. is supported by the Canada Research Chairs program (241814) and Natural Sciences and Engineering Research Council (RGPIN-2016-05159). J.C.E. acknowledges support from a NERC independent fellowship (NE/R014574/1). L.C.A is funded by the ARC ITRH for Transforming energy Infrastructure through Digital Engineering (TIDE; Grant no. IH200100009).

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
