# Peer review of "Quantifying the Uncertainty in the Eurasian Ice-Sheet Geometry at the Penultimate Glacial Maximum (Marine Isotope Stage 6)"

_The Cryosphere, 2023_

## Referee Comment (RC1)

**Review of: Quantifying the Uncertainty in the Eurasian Ice-Sheet Geometry at the Penultimate Glacial Maximum (Marine Isotope Stage 6) by Pollard *et al**

Evan J. Gowan

evangowan@gmail.com

**1  Overview**

Pollard *et al* present a reconstruction of the Eurasian Ice Sheet complex at the penultimate (MIS 6) glacial maximum. They accomplish this by using a plastic, steady state ice sheet model, ICESHEET, which creates ice sheet reconstruction using three main inputs – ice margin location, topography and basal shear stress. They develop a Bayesian statistical framework to test the ice sheet configuration by varying the basal shear stress. They calibrate the model framework by comparing with two reconstructions of the Eurasian Ice Sheet complex at the Last Glacial Maximum, namely ICE-6G and GLAC-1D. The sea level equivalent penultimate ice sheet volume after removing ruled out configurations is about 51 m, which is lower than previous dynamic ice sheet modelling exercises, but comparable to a previous result based on glacial isostatic adjustment methods.

As the creator of ICESHEET, I have been anticipating how Pollard *et al* would use my model (as a disclaimer, I did give some advice on how to run the model when the authors were starting up back in 2020). For various reasons, I personally decided against developing a Bayesian framework for my own ice sheet modelling exercises (*e.g.* Gowan et al., 2016, 2021), so I am very satisfied that Pollard *et al* have created a new way to use it. By using ICESHEET, it avoids many of the problems with other methods of ice sheet reconstruction (as described in the introduction), namely the large uncertainties in climate forcing for dynamic models, and the lack of physics in pure GIA loading models. The Bayesian framework that Pollard *et al* present provides a way forward to infer the ice sheet geometry of periods where there are few constraints on climate and sea level.

The manuscript is well written and easy to understand. Perhaps the main comment I have is that this study does not really introduce the geological observations that serve as the basis for the larger MIS 6 ice sheet compared to the last glacial cycle. For instance, we know that the ice sheet must have been much larger because the Baltic Sea was much larger in MIS 5e, even connecting to the White Sea for a time (*i.e.* Dalton et al., 2022). Having a paragraph or two introducing the geological basis would be of benefit to those interested in the ice sheet reconstruction but are not so aware of the penultimate glaciation.

**2 Comments**

**2.1 Application to current ice sheets**

When I developed my global ice sheet reconstruction (Gowan et al., 2021), I tuned the shear stress values to the present day Greenland and Antarctic Ice Sheets. Has the the Bayesian framework you developed also been applied to those ice sheets? If it is not too time consuming (*i.e.* a couple of weeks?), this would be a good test of the technique used here, as the basal shear stress can be determined directly from the present day configuration. If you anticipate such an exercise would take months, then I would regard this as optional.

**2.2 Barents Sea area at LGM**

The difficulty in fitting the LGM ice sheet in the Barents Sea with ICE-6G is not really surprising – in this area, the ICE-6G reconstruction starts with a high (and I would say unrealistic) ice thickness of nearly 5 km in the middle of the Barents Sea Ice Sheet at 26 ka. By way of comparison, the East Antarctica Ice Sheet only exceeds 4 km thickness in a few isolated places. This causes the topography in this area to be extremely depressed, which will mean that it will be harder to build up ice there with ICESHEET using realistic values of shear stress.

The result from GLAC-1D could also have issues, as we have no idea what metrics were used to tune it. GLAC-1D is an ensemble average of some unknown number of ice sheet model simulations (which we don't know because the details of the European component have never been published), and considering the likely lack of tuning parameters within the Barents Sea, could produce something that is not reflective of a real ice sheet configuration.

Though the usage of ICE-6G and GLAC-1D as a strategy to calibrate your model is fine (since they are two of the only available reconstructions of the ice sheet complex), keep in mind that these models might also not be realistic depictions of the ice sheet complex.

**2.3 Shear Stress values**

Considering that the shear stress values are the main parameter that are varied, I think it would be a good idea to include figures showing the resulting optimal shear stress values plus the associated uncertainty in the main text (right now it is only shown in the appendix).

Looking at this, I think that the ensemble values for the cold based ice shear stress values are probably set to a range that is too high. If you look at the present day Antarctica ice sheet shear stress (Fig. 1), the interior of the ice sheet tends to actually have lower shear stress values than around the margins. This is likely because the precipitation is essentially zero when the ice sheet elevation gets above 3500 m, so there is no mechanism to increase the surface gradient (and therefore increase the basal shear stress). If a similar thing happened with the penultimate Eurasian Ice Sheet complex, you would expect the shear

stress values in the middle of the ice sheet to be lower than the LGM. Using Antarctica as an analogue is not perfect, since it is an ice sheet with a relatively flat base in the interior, and mountains around the margins, which is the opposite of the Eurasian Ice Sheet Complex. Regardless, I would suspect a tendency of decreasing shear stress in the dome regions as the ice sheet grows larger than the LGM. The result of the high range (Figure 9 in the manuscript) is that the thickness of the ice sheet reaches 5 km, which is likely larger than is possible in reality.

If it is possible, I recommend running more simulations with an expanded (lower) range of Cold Based Ice Shear Stress values. I think this will result in a more realistic ice sheet configuration. The follow on to this is that I imagine the estimate of ice volume will also decrease.

**2.4   Discussion**

Perhaps a minor point, but the PGM results are compared against a reconstructed sea level curve by Waelbroeck et al. (2002) in a way that should be treated with caution. The Waelbroeck reconstruction is tuned assuming an ice volume sea level equivalent at the LGM of 130 m. The ice volume sea level equivalent will be always less than global average sea level for two reasons. First, the area of the ocean decreases as sea level falls, as continental shelf regions emerge. Secondly, the volume of the ocean basin also decreases due to GIA effects when the the water is taken out of the ocean. This means that as the ice sheets grow, it takes less ice volume to cause sea level to drop by the same amount. Estimates of sea level equivalent ice volume are therefore dependent on the choices of how to parameterize the Earth structure. The 130 m value is the result of a GIA model that is tuned against paleo sea level observations in Australia (Yokoyama et al., 2000). Australia is chosen as a good place to tune global ice sheet reconstructions, as it is a place that is expected to be close to global average sea level of around -120 m at the LGM. Recent assessments suggest the LGM sea level equivalent ice volume is likely closer to 114 m (Simms et al., 2019), a value comparable to my own analysis, which uses a different Earth model structure than Yokoyama *et al.* (Gowan et al., 2021). If this lower value is correct, which seems likely since it should be less than the near global average sea level values obtained from Australia, it will mean that the total sea level equivalent ice volume at the PGM is less than implied in the Waelbroeck curve, probably between 10-20 m.

On that note, it should be stated somewhere how the sea level equivalent ice volume is calculated (I am just guessing here that it is calculated based on modern ocean area).

**2.5   Figure 6**

I think this figure will need to be reworked in some way, or presented differently because even with my eagle-eyed vision, it is hard to see what is going on with these small plots. I think in the caption the plots need to be described better what they represent, because now it just looks like a cloud of coloured points and it is not easy to know exactly what relationship the parameters have with each other. By eye, there doesn't seem to be any clustering that would imply a relationship? Perhaps this is a consequence of the Latin Hypercube sampling, where variations happen with all parameters, so relationships between two parameters are hard to visualize? Also, I do not understand what the axes represent. Wouldn't it

be better to plot it in terms of the actual values being varied (like from Table 1)?

Increasing the font size of the other figures is also recommended.

**2.6   Code Availability**

Please ensure that any modified versions of the ICESHEET code are made available.

Best Regards,
Evan J. Gowan

**References**

**References**

Dalton, A.S., Gowan, E.J., Mangerud, J., Möller, P., Lunkka, J.P., Astakhov, V., 2022. Last interglacial sea-level proxies in the glaciated Northern Hemisphere. Earth System Science Data 14, 1447–1492.

Gowan, E.J., Tregoning, P., Purcell, A., Montillet, J.P., McClusky, S., 2016. A model of the western Laurentide Ice Sheet, using observations of glacial isostatic adjustment. Quaternary Science Reviews 139, 1–16.

Gowan, E.J., Zhang, X., Khosravi, S., Rovere, A., Stocchi, P., Hughes, A.L.C., Gyllencreutz, R., Mangerud, J., Svendsen, J., Lohmann, G., 2021. A new global ice sheet reconstruction for the past 80 000 years. Nature Communications 12, 1199.

Simms, A.R., Lisiecki, L., Gebbie, G., Whitehouse, P.L., Clark, J.F., 2019. Balancing the Last Glacial Maximum (LGM) sea-level budget. Quaternary Science Reviews 205, 143–153.

Waelbroeck, C., Labeyrie, L., Michel, E., Duplessy, J.C., McManus, J., Lambeck, K., Balbon, E., Labracherie, M., 2002. Sea-level and deep water temperature changes derived from benthic foraminifera isotopic records. Quaternary Science Reviews 21, 295–305.

Yokoyama, Y., Lambeck, K., De Deckker, P., Johnston, P., Fifield, L.K., 2000. Timing of the Last Glacial Maximum from observed sea-level minima. Nature 406, 713–716.

[Figure]

Figure 1: Present day basal shear stress values for Antarctica in PaleoMIST (Gowan et al., 2021)

---

## Author Comment (AC2)

**Response to the Reviewers**

Colour Key
Reviewer Comment, Author Response, Enacted Manuscript Change

We would like to thank reviewer 1 (Dr Gowan) and reviewer 2 (Prof Tarasov) for their valuable feedback on our manuscript. The reviews have highlighted areas where our experimental design and statistical methodology could be improved, and we have since performed a new ensemble experiment for the PGM with 1000 members that includes parameters controlling the margin extent and magnitude of topographic deformation. In addition, we modify our implausibility calculation for the last deglaciation to include estimates for model structural error and model bias. We include the use of emulation of the ice-sheet volume estimates to enable a more thorough exploration of the implausibility space, ensuring to include the emulator variance in our implausibility, and use this methodology to filter our new PGM ensemble. Overall our results do not change significantly, with former our PGM volume reconstruction of 51+-6m, updated to 49+-8 m taking in to account our wider assessment of the uncertainties. Below is a new figure summarising the result.

[Figure]

*Fig A (Volume Probability Distributions): Figure shows the probability distribution of total ice-sheet volume for the Eurasian ice sheet based on the ICESHEET ensemble before (bottom) and after (top) applying the history matching constraints for the GLAC-1D 20ka (blue), ICE-6G 22ka (green), and ensemble of possible PGM margin extents (red).*

**Reviewer 1: Evan Gowan**

1 Overview

Pollard et al present a reconstruction of the Eurasian Ice Sheet complex at the penultimate (MIS 6) glacial maximum. They accomplish this by using a plastic, steady state ice sheet model, ICESHEET, which creates ice sheet reconstruction using three main inputs – ice margin location, topography and basal shear stress. They develop a Bayesian statistical

framework to test the ice sheet configuration by varying the basal shear stress. They calibrate the model framework by comparing with two reconstructions of the Eurasian Ice Sheet complex at the Last Glacial Maximum, namely ICE-6G and GLAC-1D. The sea level equivalent penultimate ice sheet volume after removing ruled out configurations is about 51 m, which is lower than previous dynamic ice sheet modelling exercises, but comparable to a previous result based on glacial isostatic adjustment methods.

As the creator of ICESHEET, I have been anticipating how Pollard et al would use my model (as a disclaimer, I did give some advice on how to run the model when the authors were starting up back in 2020). For various reasons, I personally decided against developing a Bayesian framework for my own ice sheet modelling exercises (e.g. Gowan et al., 2016, 2021), so I am very satisfied that Pollard et al have created a new way to use it. By using ICESHEET, it avoids many of the problems with other methods of ice sheet reconstruction (as described in the introduction), namely the large uncertainties in climate forcing for dynamic models, and the lack of physics in pure GIA loading models. The Bayesian framework that Pollard et al present provides a way forward to infer the ice sheet geometry of periods where there are few constraints on climate and sea level.

The manuscript is well written and easy to understand. Perhaps the main comment I have is that this study does not really introduce the geological observations that serve as the basis for the larger MIS 6 ice sheet compared to the last glacial cycle. For instance, we know that the ice sheet must have been much larger because the Baltic Sea was much larger in MIS 5e, even connecting to the White Sea for a time (i.e. Dalton et al., 2022). Having a paragraph or two introducing the geological basis would be of benefit to those interested in the ice sheet reconstruction but are not so aware of the penultimate glaciation.

We thank the reviewer for their supportive summary of our work and for their assessment that the outlined framework provides a useful methodology for tackling poorly constrained ice-sheet geometries. We agree with their comment that the geological evidence is not sufficiently highlighted in the work.

Additional paragraph added to introduction highlighting geological basis for larger MIS 6 Eurasian ice sheet.

**2 Comments**

**2.1 Application to current ice sheets**

When I developed my global ice sheet reconstruction (Gowan et al., 2021), I tuned the shear stress values to the present day Greenland and Antarctic Ice Sheets. Has the the Bayesian framework you developed also been applied to those ice sheets? If it is not too time consuming (i.e. a couple of weeks?), this would be a good test of the technique used here, as the basal shear stress can be determined directly from the present day configuration. If you anticipate such an exercise would take months, then I would regard this as optional.

We thank the reviewer for the suggestion of expanding the techniques applied here to other ice sheets in order to improve validation. We agree that the framework could be extended to include present-day configuration of ice sheets, but that limited information would be gained for comparison to the PGM. This is because we envision that the abundance of present-day data would create a highly refined parameter set that is not necessarily appropriate for the PGM.

No change, given the focus of the paper is reconstructing the Eurasian PGM maximum.

**2.2 Barents Sea area at LGM**

The difficulty in fitting the LGM ice sheet in the Barents Sea with ICE-6G is not really surprising – in this area, the ICE-6G reconstruction starts with a high (and I would say unrealistic) ice thickness of nearly 5 km in the middle of the Barents Sea Ice Sheet at 26 ka. By way of comparison, the East Antarctica Ice Sheet only exceeds 4 km thickness in a few isolated places. This causes the topography in this area to be extremely depressed, which will mean that it will be harder to build up ice there with ICESHEET using realistic values of shear stress. The result from GLAC-1D could also have issues, as we have no idea what metrics were used to tune it. GLAC-1D is an ensemble average of some unknown number of ice sheet model simulations (which we don't know because the details of the European component have never been published), and considering the likely lack of tuning parameters within the Barents Sea, could produce something that is not reflective of a real ice sheet configuration. Though the usage of ICE-6G and GLAC-1D as a strategy to calibrate your model is fine (since they are two of the only available reconstructions of the ice sheet complex), keep in mind that these models might also not be realistic depictions of the ice sheet complex.

We agree with the reviewer's assessment that ICE-6G and GLAC-1D are both likely to be flawed in some capacity. We further address the issue of consistently poorly matched regions, such as the Barents Kara ice sheet in ICE-6G, through the introduction of a bias term in the implausibility procedure (also in response to comments by reviewer 2). This term can be viewed as an expression of acceptable mismatch between ensemble members and the target reconstruction. GLAC-1D and ICE-6G are the only suitable and available models for this time period that are also independent of ICESHEET. However, should further models become available in future, we believe the framework outlined in this paper could easily be expanded to include the additional constraint data. We also wish to highlight the usefulness of uncertainty estimates on target reconstructions, should they be provided.

We introduce a model bias correction field to mitigate issues surrounding consistently poorly represented regions.

**2.3 Shear Stress values**

Considering that the shear stress values are the main parameter that are varied, I think it would be a good idea to include figures showing the resulting optimal shear stress values plus the associated uncertainty in the main text (right now it is only shown in the appendix).

Figure 6 shows updated distributions from the shear stress parameters post history matching. However, it doesn't give an indication of the best values, or what this optimal configuration looks like spatially. In order to better display the implications of the history matching procedure, we have chosen to train a gaussian process emulator to be able to rapidly predict the implausibility value for any given shear stress combination. Utilising this tool, we have performed 33.6 million samples to visualise the optical depth of the parameter space, showing regions with more accepted members. We are hesitant to include an 'optimal' shear stress combination, as our framework is designed to rule out implausible combinations of parameter values rather than optimum values. Instead, we have combined the mean NROY shear stress map with the new optical depth figure, replacing figure 6.

[Figure]

*Fig B (NROY Optical Depth): The density of the NROY space (reds) and the minimum implausibility value (greens) shows for each face of the 7-dimensional hypercube. Each panel is composed of 40x40 cells, while the value at each cell is derived from a 1000 member random sample of 12 gaussian process volume emulators which are used to calculate the resulting implausibility.*

Replaced figure 6 with new figure visualising the optimal parameter space regions via an optical depth figure of the parameter space, and included shear stress map of mean NROY inputs.

Looking at this, I think that the ensemble values for the cold based ice shear stress values are probably set to a range that is too high. If you look at the present day Antarctica ice sheet shear stress (Fig. 1), the interior of the ice sheet tends to actually have lower shear stress values than around the margins. This is likely because the precipitation is essentially zero when the ice sheet elevation gets above 3500 m, so there is no mechanism to increase the surface gradient (and therefore increase the basal shear stress). If a similar thing happened with the penultimate Eurasian Ice Sheet complex, you would expect the shear stress values in the middle of the ice sheet to be lower than the LGM. Using Antarctica as an analogue is not perfect, since it is an ice sheet with a relatively flat base in the interior, and mountains around the margins, which is the opposite of the Eurasian Ice Sheet Complex. Regardless, I would suspect a tendency of decreasing shear stress in the dome regions as the ice sheet grows larger than the LGM. The result of the high range (Figure 9 in the

manuscript) is that the thickness of the ice sheet reaches 5 km, which is likely larger than is possible in reality. If it is possible, I recommend running more simulations with an expanded (lower) range of Cold Based Ice Shear Stress values. I think this will result in a more realistic ice sheet configuration. The follow on to this is that I imagine the estimate of ice volume will also decrease.

We agree with the suggestion that the cold based ice shear stress range may exceed realistic values in the initial ensemble design. However, we find little corelation between the prescribed cold ice shear stress value and the maximum ice thickness. Instead, the strongest control is the value of the cold ice interior distance, as a larger cold ice region will allow ICESHEET the space to generate a higher ice dome. The extreme values of the interior distance are filtered out after history matching, reducing the prevalence of > 5km ice thicknesses. However, despite history matching reducing our mean maximum thickness from 5.100 +- 0.8 km to 4.6 +- 0.7 km, some simulations with high thickness values remain. If we apply a filter that removes simulations with higher than 5km maximum thickness values, we reduce the mean volume from 49 +- 8 m (SLE) to 46 +- 7m SLE, and mean maximum thickness to 4.2 +- 0.5 km. We will include a discussion around this constraint, alongside a figure showing the alternative probability distribution, in the main text.

[Figure]

*Fig C (Constraining based on maximum thickness): Figure showing the total volume probability distributions for the PGM Eurasian ice sheet in the original ensemble (blue), post-history matching (orange), and after removing simulations with a maximum thickness of 5km or more.*

Investigated the impact of constraining the maximum thickness on the total volume distribution, and will include a summary of the findings and accompanying figure in the discussion section.

2.4 Discussion

Perhaps a minor point, but the PGM results are compared against a reconstructed sea level curve by Waelbroeck et al. (2002) in a way that should be treated with caution. The Waelbroeck reconstruction is tuned assuming an ice volume sea level equivalent at the LGM of 130 m. The ice volume sea level equivalent will be always less than global average sea level for two reasons. First, the area of the ocean decreases as sea level falls, as continental shelf regions emerge. Secondly, the volume of the ocean basin also decreases due to GIA effects when the the water is taken out of the ocean. This means that as the ice sheets grow, it takes less ice volume to cause sea level to drop by the same amount. Estimates of sea level equivalent ice volume are therefore dependent on the choices of how to parameterize the Earth structure. The 130 m value is the result of a GIA model that is tuned against paleo sea level observations in Australia (Yokoyama et al., 2000). Australia is chosen as a good place to tune global ice sheet reconstructions, as it is a place that is expected to be close to global average sea level of around -120 m at the LGM. Recent assessments suggest the LGM sea level equivalent ice volume is likely closer to 114 m (Simms et al., 2019), a value comparable to my own analysis, which uses a different Earth model structure than Yokoyama et al. (Gowan et al., 2021). If this lower value is correct, which seems likely since it should be less than the near global average sea level values obtained from Australia, it will mean that the total sea level equivalent ice volume at the PGM is less than implied in the Waelbroeck curve, probably between 10-20 m.

On that note, it should be stated somewhere how the sea level equivalent ice volume is calculated (I am just guessing here that it is calculated based on modern ocean area).
We thank the reviewer for their detailed insight into the assumptions behind the use of the Waelbroeck curve and modify the text in order to highlight these points. We also make sure to state how sea-level equivalent ice volume is calculated in this work (which, as suggested, is just based on modern ocean area).
We include points of relevance in the discussion section, and included SLE calculation description.

2.5 Figure 6

I think this figure will need to be reworked in some way, or presented differently because even with my eagle-eyed vision, it is hard to see what is going on with these small plots. I think in the caption the plots need to be described better what they represent, because now it just looks like a cloud of coloured points and it is not easy to know exactly what relationship the parameters have with each other. By eye, there doesn't seem to be any clustering that would imply a relationship? Perhaps this is a consequence of the Latin 4Hypercube sampling, where variations happen with all parameters, so relationships between two parameters are hard to visualize? Also, I do not understand what the axes represent. Wouldn't it be better to plot it in terms of the actual values being varied (like from Table 1)?

Increasing the font size of the other figures is also recommended.
Figure 6 is now revamped with the use of gaussian process emulation to explore the optical depth of the parameter space (as per comment 2.3). This is presented in terms of actual parameter values, rather than the unit scaled values, and should hopefully be much clearer.

New figure for figure 6 developed, and figure font sizes increased. We also change the colour scales as requested by the journal.

2.6 Code Availability

Please ensure that any modified versions of the ICESHEET code are made available.

The ICESHEET modified code is available on GitHub and will be linked to in the manuscript. Perhaps there is a way to add this as a branch to the original ICESHEET repo? I will contact the reviewer (as the developer of ICESHEET) to discuss further.

The ICESHEET github code linked to in-code availability statement.

**Reviewer 2: Lev Tarasov**

(note my quality/impact/... ratings on the review form are based on the current version)
The Pollard et al submission is an attempt of applying history matching to the Penultimate Glacial Maximum (PGM) for Eurasia. The choice of history matching is appropriate, however the implementation is limited and currently inadequate to match the claims. Critical, the submissions claims via the title to quantify "the Uncertainty in the Eurasian Ice-Sheet Geometry at the Penultimate Glacial Maximum", and via the abstract to "robustly quantify uncertainties" and yet it only partially does so.

We thank reviewer 2 for their endorsement of our decision to employ history matching in this work. In light of the reviewer's comments, we have significantly expanded our PGM ensemble size to include 1000 members. This new ensemble also incorporates two additional parameters to account for uncertainty in the topographic deformation and ice extent. Furthermore, we have introduced 12 gaussian process emulators, and redesigned our implausibility metric to include model structural error and bias estimates as suggested by the reviewer, in order to more accurately define the NROY space. Finally, emulation is used to more concretely express the probability distribution of the PGM Eurasian ice sheet volume. After making these extensive revisions, we find that our final mean PGM Eurasian volume result is modified by less than 5%, from 51+-6m to 49+-8 m.

Given the reviewers further comments regarding uncertainty quantification, which we address in turn below, we believe that we now do quantify the uncertainty (where appropriate), and this remains a valid title. The word 'robustly' is not present in the abstract and therefore no edit required.
No change

It fails to account for the structural uncertainty of their static perfectly plastic ice sheet model. It also fails to account for uncertainties in the MIS 6 ice margin. A key point is that MIS 6 maximum ice extent has poor age control. It is unclear to what extent parts of the margin represent short term surge events (which are not going to be well represented by a perfectly plastic ice sheet model), nor is it clear to what extent the maximum ice margin extent was synchronous. Nor is the potentially large uncertainty associated with the assumption of an equilibrium ice sheet addressed.

We agree with the reviewer's comment that some sources of uncertainty they identify were not initially included in the modelling work, despite this already being a very comprehensive study. An estimate of model structural uncertainty is now represented in the implausibility metric (as detailed below) with the inclusion of a model bias term as suggested by the reviewer (detailed below). It follows that penultimate glacial maximum reconstructions may also contain this bias term, and we account for this by calculating the 22ka average percentage total volume bias, and applying this to the PGM volume distribution via the mean PGM volume. Estimation of the model bias field that would be required to display a debiased geometry would be highly subjective and we instead choose to display our most probable PGM geometry as the ensemble member closest to our mean de-biased PGM volume estimate.

To address the margin uncertainty, we have completely rerun the PGM ensemble, increasing the ensemble size to 1000 members, and including an additional parameter that controls the extent of the ice sheet margin. We believe the MIS 8 best estimate margin, and MIS 6

maximum margin from Batchelor et al. (2019) represent reasonable estimates on the plausible minimum/maximum margin extents respectively (considering the limited geological constraints available). Our new margin parameter is a continuous scalar value where 0 represents the minimum (for which we use the Batchelor MIS 8 margin, given uncertainties on timing of the advances of the Saalian-complex chronologies in Eurasia), 0.5 represents the Batchelor (2019) MIS 6 best estimate margin (as used in the original manuscript), and 1 represents the maximum MIS 6 margin extent. Margins corresponding to any values between 0 and 1 are generated by linear interpolation between respective margins, using a novel shape interpolation algorithm we have developed. However, we agree that this does not resolve the issue of synchroneity of extent for any margin tested and we therefore highlight that our approach is limited to assuming a synchronous maximum margin.

While we believe the assumption of an equilibrium ice sheet is acceptable during the PGM, we agree that this could be better explained in the text, and include additional text to highlight this.
We have run a new ensemble for the PGM (1000 members), with 2 additional parameters, once for the margin, and another for topography (as described below). We include addition information in the text describing these new parameters, and highlight that we are limited to assuming a single synchronous margin with an ice sheet assumed to be at equilibrium.

The work is of potential value, but it first needs to make claims that are defensible. This includes clarity and accuracy on the extent to which uncertainties are addressed and that this is a very approximate history matching as the parameter sampling is far from complete. History matching typically relies on emulators to adequately sample the parameter space. 200 samples for 7 parameters is far from adequate unless the response is very linear with minimal interaction between parameters (which would have to be shown).
We thank the reviewer for their suggestion of improved history matching with the use of emulation. In our original experimental design, we did not require the use of emulation to make use of the history matching procedure, since the same parameter sample was used for both the Last Deglaciation and PGM runs, meaning that no interpolation was required and so the implausibility had no emulator uncertainty at these points for both sets of simulations. However, we agree with the reviewer that to better understand the shape of the NROY space, and now to map implausibility to our new PGM sample, requires the use of emulation. Accordingly, we have built a set of 12 Gaussian process emulators to emulate each ice sheet volume metric during the last deglaciation. The predicted volumes from the emulators are used to calculate implausibility (now including a term for emulator variance) for arbitrary parameter samples. We employ this methodology to produce an optical depth plot of the parameter space, which replaces figure 6, (and is shown below), as well as to rule out members of the new PGM sample. We also update the manuscript with a new section to reflect these changes.
We develop GP emulators trained on volume metrics, enabling better assessment of the implausibility space beyond explicitly modelled sample values. A new section is added to the text to reflect this addition to the manuscript.

Secondly, the chosen model uncertainty estimate has no justification. Futhermore, it is clear that there is a bias error that also needs to be addressed in the implausibility function given

the mean NROY misfits to the GLAC1D and ICE6G ice sheet chronologies. However, this can be rectified. Select at least 20 of your simulations that have the least RMSE error for GLAC1D (and separately for ICE6G if that is fully from a glaciological model). Use the variance of the residuals as your minimum structural model variance error estimate and use the mean bias as your minimum model bias error estimate in the implausibility. Note, these values will clearly vary around the ice sheet. You can either make the error estimates a field (ie depending on relative location in the ice sheet), or you can choose the maximum value across the ice sheet (easier but at cost of wider uncertainties). As both of these ice chronologies have their own limitations, expanding the resultant variance and bias estimates by some fudge factor, would still be needed.

We thank the reviewer for their insight and helpful suggestions for ways to improve our assessment of model structural error, and for highlighting the need to account for model bias. We have replaced our original estimate of model structural error, and included a model bias correction, with the method provided in the reviewer's comment. We calculated the residuals of the 20 simulations with the lowest RMSE per time and model, and converted these into regional volume bias and structural error estimates, which were then included in the implausibility. In addition, we choose to expand the variance estimate by a fudge factor, F, of 1.2. Our updated implausibility equation is below:

$$I^2(p) = \frac{\left(\mathbb{E}(O) - (\mathbb{E}(M(p)) - M_{bias})\right)^2}{\left(Var(O) + Var(M)\right)F + Var(GP)}$$

Overhauled implausibility calculation, included in manuscript.

To account for errors growing with a larger PGM ice sheet, scale the uncertainties for ice thickness by the ratio of mean ice height.

We scale the model bias term to account for this issue in the volume distribution estimate. Scaled bias correction based on PGM volume distribution.

Thirdly, the authors are conflating NROY with plausible. This is a problematic stretch. Showing something as being NROY, ie not implausible, doesn't necessarily make it plausible.

We agree with the reviewer and have clarified this where appropriate within the text. Modified text to correct for this.

Finally, the authors do not adequately address the limitations of the perfectly plastic approximation and make a number of inaccurate claims, some of which are detailed below (running late on this review, I figured it's better to give you something to work with now than a completely detailed evaluation).
**some specific comments**

Quantifying the Uncertainty in the Eurasian Ice-Sheet Geometry at the Penultimate Glacial Maximum (Marine Isotope Stage 6)
**The title is misleading, as uncertainties are inadequately quantified.**

We refute the claim that the title is misleading. The geometry of the Penultimate Glacial Maximum ice sheet is a highly uncertain quantity. Throughout this work, we quantify the magnitude of this uncertainty by accounting for the dominant contributions present within the specific numerical modelling procedure applied here. It is true that other, smaller sources of uncertainty may exist that are not explicitly accounted for through this work, but it is unreasonable to assume that the title of a publication implies a complete, allencompassing assessment of the topic at hand (see first comment), and with our improved experimental design we feel this title is now even more appropriate.
No change.

51.16±6.13 m sea-level equivalent
**The significant digits given for results are meaningless, correct this to an appropriate amount.**
We correct the significant figures throughout the text.
Values reduced to appropriate sig fig e.g. 51 ± 6m.

We perform Bayesian uncertainty quantification
**History matching is not Bayesian (where do you invoke Bayes Rule?)**
**cf https://www.physics.mun.ca/~lev/revCalG.pdf**
**for an explanation of what Bayesian is and entails.**
We disagree. The central tenet in Bayesian statistics is the subjectivist viewpoint of probability. Bayes theorem (which results from probability theory and can be shown from laws of conditional probability) has a particular interpretation in Bayesian statistics as the revision of subjective probability under new evidence. To be Bayesian does not require one to evoke Bayes theorem, but simply to view probability as a degree of belief. Regardless of these arguments, in this version of the manuscript we have included Gaussian process emulation into the history matching procedure and so our methods are distinctly Bayesian. The reference to our work as Bayesian is not only appropriate, but scientifically useful and we do not feel the need to revise our use of the term.
No Change.

Finally, the simple ice-sheet model approach is designed to generate ice geometries based on simple, steady-state ice-sheet physics for a prescribed margin
**Misleading as stated. The perfectly plastic ice sheet model is derived from "steady-state ice-sheet physics" but it doesn't reflect it, only provides a limited approximation.**
To more accurately describe the model, we replace the text with "the simple ice-sheet model approach is designed to generate ice geometries that approximate the profile of a steady-state ice-sheet for a prescribed margin".
Rephrasing of highlighted statement in the text.

Simple ice-sheet model whose minimal input requirements
**The specification of a 2D basal shear stress map for each timeslice is far from minimal.**
The decomposition of the basal shear stress map parameterises this input. It is usual to refer to a field input to a model as a single input. A dynamic ice-sheet models that, for example, take 32 inputs, one of which is an initial topography configuration on a 1000x1000 cell grid, is not referred to as being a 1000031 parameter model. In this context, while the 2D basal shear stress map represents significant degrees of freedom, which are addressed through regional sediment categorisation, we believe the phrase minimal is still appropriate.
No change.

However, reliance on poorly constrained rebound data required for GIA inversion modelling (Lambeck et al., 2006) or assumptions of highly uncertain climate data used in dynamic icesheet simulations (Abe-Ouchi et al., 2007; Peyaud, 2006) make these approaches challenging to constrain for the Penultimate Deglaciation and give only a very limited view of possible pasts with no grasp on the vast range of plausibility

**The last claim is incorrect. I'm using a glaciological (hybrid shallow shelf/shallow ice) model with large degrees of freedom in the climate forcing. Though not perfect, I suspect I've generated a larger range of history matched simulations for last glacial cycle Eurasian ice sheet evolution (in early process of write-up) than is offered by the methodology of this submission, as evidenced by my current 50 ensemble parameters (versus 7 in this submission).**

Thank you for highlighting your work in progress. We look forward to reading the manuscirpt when it is published. Unfortunately, the editorial rules restricts the authors to only citing current peer-reviewed, published work.

No change.

A simple ice-sheet model whose minimal input requirements enables the production of large ensemble simulations with controlled sources of uncertainty (Gowan et al., 2016a).

**If the sources of uncertainty are "controlled", then they should be fully assessed.**

We are unsure of the meaning of this comment. Perhaps we can reiterate here, that in response to other comments by the reviewer, we have now included: parameters to account for additional sources of topographic deformation and margin extent uncertainty; reformulated the implausibility metric calculation; employed gaussian process emulation to extend our ability to explore the NROY space; ran a new, 1000 member ensemble for the penultimate glacial maximum; and modified language (where appropriate) to increase transparency in our methodology. In light of these modifications, we believe the uncertainties are now well assessed.

No change

generate physically plausible ice-sheet reconstructions

**Given the approximations involved, I don't see how you can call these physically plausible.**

The ice-sheet reconstructions are physically plausible within the context of the assumptions outlined in detail.

No change

since our simulations are process based

**The perfectly plastic approximation is not a process based model but a diagnostic tool. What processes are you actually modelling?**

We agree with the reviewer that the term process is misleading, given its connotation with dynamics. We modify the text to remove the mention of process and instead refer to the modification to the basal shear stress map in a way that does not imply dynamics.

Removed the mention of processes in reference to ICESHEET and the shear stress map modification in the text.

The model has been successfully applied where large uncertainty in inputs required for dynamic ice-sheet models, such as climate, have reduced the confidence in using the outputs of such models as inputs to sea-level models due to misfits against ice extent and volume distributions that impact GIA, and where large numbers of runs are required making

computation efficiency 105 paramount, such as in the exploration of variable global ice-sheet configurations (Gowan et al., 2021)

**What do you mean by "successfully"? Instead of vague descriptors, be precise or do you want to be stuck with my definition of "successful"?**

We agree with the reviewer that the term 'successfully' is vague and remove it in this instance.

Removed the term 'successfully' from this sentence.

Limited constraints on climatic conditions, the requirement for large ensemble simulations to explore the range of plausible scenarios, and a need for well-defined sources of uncertainty make ICESHEET an ideal choice for exploring uncertainty in ice sheet configurations during the PGM.

**Again, given the statement above, a full uncertainty assessment should be provided to match the claim. Without seeing that assessment, I see no basis for calling ICESHEET an "ideal" choice, or even a defensible choice.**

We believe we have defended the choice of ICESHEET in the text as being the optimal choice within our criteria for this work: it is able to generate ice-sheet loads that are based on simple physics, it is computationally cheap such that large ensemble experiments are easily facilitated, and it does not require the prescription of large numbers of parameters required for climate-driven dynamic ice sheet models. ICESHEET has also been applied to other, similar challenges (Bradley et al., 2023). In the context of generating ice-sheet load inputs to GIA models (which is the purpose of this work) we believe that ICESHEET is indeed the ideal choice.

No change

In order to account for GIA, we assume that the Eurasian ice sheet at the PGM had been at its maximum extent sufficiently long for the solid Earth underneath to be at (or close 180 to) an isostatic equilibrium with the ice load, an assumption we consider reasonable given the lack of constraints during this time

**this is a large assumption, contributing another source of unassessed uncertainty. One main reason for your lack of constraints, is your missing of full glaciological physics of the ice and earth system which provide some pretty strong constraints on the system.**

We agree with the reviewer that assumptions on fully relaxed topography may contribute some uncertainty to the output, but in initial testing we found this to be a small contribution. However, we have decided to include a new parameter in the experimental design, topo_equilib, which can vary between 0 and 1 and that scales the equilibrium deformation field, thus reducing the size of the depression for a given reconstruction. Since the assumption of modern-day topography (topo_equilib = 0) is clearly wrong, we set the lower bound value of our parameter to be equal to the value which best matches the GLAC-1D 22ka topography, given the GLAC-1D 22ka ice sheet load, using our method, resulting in around a 0.46 minimum parameter value. We find that this parameter contributes less than 10% uncertainty to the ice sheet volume.

New PGM ensemble ran with new parameter to include first order assessment on the contribution of uncertainty due to topographic equilibrium.

We run 200 simulations for each reconstruction and each of the 4 selected time periods (22, 20, 18, 16 ka), totalling 1600 simulations (Figure 5 and Figure A1).

**so you are assuming minimal interactions between your parameters, but do not provide evidence to support this. If this were not the case, then even just a 3 value min/median/max grid search over 7 parameters would entail 3^7= 6561 simulations for 1 timeslice.**

We believe the reviewer is referring to the fact that, even with a simplistic one-at-a-time parameter perturbation experiment, a parameter space with 7 dimensions requires thousands of simulations to explore properly. We disagree that such large ensembles are required to assess the impact of interactions, and that interactions do not need to be quantified for the purposes of history matching. The maxi-min Latin Hypercube sampling used in our study is a highly efficient procedure that is routinely used to explore multi-dimensional parameter space for history matching. As a rule of thumb, it is customary to have an ensemble size that is a least 10 times the number of parameters varied.

No change

eq 1

**The denominator for implausibility should include variance for model structural error. The numerator should include model structural bias error. You are choosing to specify model structure error as simply some fraction of ensemble variance. On what basis do you justify such a choice?  To understand why this can be problematic, cf the simple example in subsection 2.5 of https://doi.org/10.5194/egusphere-2022-1410. The above reference also provides guidance some guidance on how model structural uncertainty can be defensibly assessed.**

We thank the reviewer for their insight here and, as stated above, we have modified our implausibility procedure. The new equation now includes model structural error and model bias error.

Implausibility equation changed to include new terms and remove original structural error assessment.

Bayesian History Matching
**Looking at the first page of googled hits for "Bayesian History Matching", the actual papers describe Bayesian emulation used in history matching. As you are not using emulators, your description is inaccurate. This needs to be corrected throughout.**

We disagree with the reviewers comment that Bayesian history matching requires the use of emulation. History matching is a Bayesian calibration tool used to rule out regions in a parameter space. Historically, emulators have been used in history matching to interpolate between model runs over the parameter space (Williamson et al., 2013). In our original manuscript we were only interested in calculating implausibility at the observed locations in parameter space and so did not require emulation. As described above, we have since included emulation into the paper's workflow and so do not feel the need to rectify our use of the term.

No change to phrasing in light of revisions to the model design.

We restrict our NROY space to parameter values that correspond to models runs with implausibility $I(\hat{p})$ higher than 3, following the Pukelsheim (2012) three-sigma rule typically used in Bayesian History Matching
**Should be less than 3 not "greater than 3" for NROY. You should also state clearly what assumptions you are making about the residual distribution to justify the choice 3 sigma (the**

modelling community relies too much on "this is what others do"). Cf the above https link for the assumptions made.

Typo has been corrected to less than 3. The three-sigma rule, commonly used in history matching, can either be derived from Chebyshev's or Pukelsheim's inequalities. Chebyshev makes no assumptions on the distributional form and his inequality is simply a result of simple probability theory, showing that at minimum 89% of the distributional mass will lie within 3 standard deviations of the mean. Pukelsheim revises Chebyshev's inequality for unimodel Lebesgue distributions and tightens these bounds to a minimum of 95% of the distributional mass lying between 3 standard deviations. It would be highly unreasonable to assume that our residual density is not Lebesgue measurable (such examples are confined to esoteric corners of measure theory) and it is up to you as modellers to defend the assumption of unimodality. The choice of 3 as a limit is arbitrary, as are many decisions in statistics (and much of science), however we clearly document our choices here. It should be noted that the choice of 3 standard deviations is a massive bound as compared to most chosen analytical distributional forms (certainly the exponential family of distributions; e.g. normal, exponential, t...). So while we could choose something other than 3, given the very general assumptions behind these bounds we do not feel the need to deviate from the views of the overwhelming majority of the statistical community that are happy with the choice of 3.

Figure 5

**please add a few contours to each frame as its hard to discern the colour map values within even 300 m Also, I can't make sense of the colour scale for the first two columns. How can you have negative ice thickness or ice surface elevation (not clear what is being plotted) a km below sealevel?**

We agree that these figures could be clearer, and have modified them to ensure legibility of values.

Thickness maps modified for legibility.

Our work has expanded this methodology to include the cold-based ice and active ice streaming basal processes which have had a strong impact on the implausibility metric, improving the simulation fit during history matching when applied to the Last Deglaciation, with the exception of the British ice sheet (Figure 4) where simulation mismatch is likely due to discrepancies in ice-margin extraction.

**You need to be honest about exactly what the perfectly plastic approximation entails and how far what you implement is from the physics of ice streams.**

We agree that the notion of processes should be changed to be more transparent in our methodology. These changes should remove issues surrounding suggestions of implementation of ice stream physics.

Process as a descriptor removed.

---

## Author Response (AR2)

**Response to the Editor**

Colour Key
Reviewer Comment, Author Response, Enacted Manuscript Change

We would like to thank the editor for reviewing our manuscript and for making helpful suggestions and corrections to further improve the clarity of our work.

**Editor Comments**

Public justification (visible to the public if the article is accepted and published):
To the authors,

You have done a thorough job responding to the reviewers' comments. While several criticisms of the methodology arose from the reviews, you convincingly responded as well as adapted the methodology where needed. Minor revisions are needed, as outlined below, before the manuscript can be published.

Best regards,
Alex

Minor comments:

L1: North Sea => The North Sea
Corrected

L5: sheer => shear
Corrected

L19: as well as those => as well as by those
Corrected

L32: focus upon => focus on
Corrected

L32: during LIG => during the LIG
Corrected

L72: Delete "by"
Corrected

L86: dynamical => dynamic
Corrected

L98: sheer => shear
Corrected

L99: ensemble => an ensemble
Corrected

L99: sheer => shear [and throughout the text]
Corrected

L112: sheer
Corrected

L115: computation => computational
Corrected

L162: accuracy => accurately
Corrected

L174: upon late => on the late
Corrected

L275: In Figure 2, only three categories are given in panel A, which I believe is the relevant panel here. Perhaps you could add some contours to indicate the independent categorized regions, or explain how they relate to what is shown in the map.
We thank the editor for highlighting this. The original phrasing of this paragraph is confusing. For clarification: Panel A shows only three categories as we don't use the original 5 category map that we describe in the text. We instead use a modified version that has simplified two of these categories into one (three categories, panel A), while also separated out the ice streaming category (panel B) to allow for modification to the extent of ice streaming. This is so if we shrink the extent of ice streaming, we still have a shear stress value underneath to use, whereas the previous map had the ice streaming regions burned in.
We believe that our changes to the text should now clarify this point, and believe that modifications to the figure to include additional contours may act to confuse things slightly, and so we choose to leave the figure as is, but we have modified the figure caption to improve clarity.
Clarified that the original map is a separate piece of work, and merged paragraph to highlight that we used a modified version. Explicitly stated and referenced panels A and B for the 3 sediment categories that underly the map and ice streaming categories respectively.

L279: dynamical => dynamic
Corrected

L403: "less than 3" <= I think what you mean here is "greater than 3". The NROY space you define is less than 3 (as mentioned earlier and pointed out by the reviewer), but here you remove simulations that have values greater than 3. Please check and clarify. As it is written, "less than 3" does not make sense here.
Thank you for correcting this typo, you are correct in that we meant to exclude members with greater than 3 sigma implausibility. This got mixed up when correcting the previous typo throughout the text.

Corrected

L417: is likely reflecting => likely reflects
Corrected

L418: rain show reductions => rain shower reductions [? This phrase is not clear.]
Here we meant the effect of a rain shadow on accumulation.
Corrected typo and clarified sentence in text.

L419: ice-dome dome => ice dome
Corrected

L420: consistent => is consistent
Corrected

L564: "By obtaining ..." <= This sentence is not clear, please revise.
Here we meant that having access to the full ensemble from previous works, rather than just the mean/single ensemble member, would allow for more robust analysis of observation uncertainties.
Changed sentence to clarify this point.